# Simulating Human-like Daily Activities with Desire-driven Autonomy

**Yiding Wang** [*◇], **Yuxuan Chen**[*♡], **Fangwei Zhong**[✉♠†], **Long Ma**[♣†], **Yizhou Wang**[◊◇]

◇ Institute for Artificial Intelligence, Peking University ♡ The University of Hong Kong

♠ School of Artificial Intelligence, Beijing Normal University

♣ Academy for Advanced Interdisciplinary Studies, Peking University

† State Key Laboratory of General Artificial Intelligence, BIGAI

◊ Center on Frontiers of Computing Studies, School of Computer Science,

Nat'l Eng. Research Center of Visual Technology, Peking University

* indicates equal contribution ✉Correspondence to `fangweizhong@bnu.edu.cn`

## Abstract

Desires motivate humans to interact autonomously with the complex world. In contrast, current AI agents require explicit task specifications, such as instructions or reward functions, which constrain their autonomy and behavioral diversity. In this paper, we introduce a Desire-driven Autonomous Agent (D2A) that can enable a large language model (LLM) to autonomously propose and select tasks, motivated by satisfying its multi-dimensional desires. Specifically, the motivational framework of D2A is mainly constructed by a dynamic *Value System*, inspired by the Theory of Needs. It incorporates an understanding of human-like desires, such as the need for social interaction, personal fulfillment, and self-care. At each step, the agent evaluates the value of its current state, proposes a set of candidate activities, and selects the one that best aligns with its intrinsic motivations. We conduct experiments on Concordia, a text-based simulator, to demonstrate that our agent generates coherent, contextually relevant daily activities while exhibiting variability and adaptability similar to human behavior. A comparative analysis with other LLM-based agents demonstrates that our approach significantly enhances the rationality of the simulated activities [1].

## 1 Introduction

The quest to create human-like autonomous agents (Wang et al., 2016; 2024b) has long been a prominent theme in artificial intelligence (AI) (Fetzer, 1990). A human-like autonomous intelligence is not simply a task executor; it can generate and execute tasks within a motivational framework aligned with its values and goals. Early pioneers such as Alan Turing laid the groundwork for evaluating intelligence through frameworks such as the Turing Test (Pinar Saygin et al., 2000), which posited that a key measure of an agent's intelligence is the ability to produce behavior indistinguishable from that of a human. Consistent with this perspective, we believe that generating controllable and reasonable human-like behavior is crucial to agents. Such capabilities can also enhance user engagement and trust in various applications of AI agents, from service-oriented intelligent assistants that foster realistic interactions (Vrba et al., 2014) to virtual agents in games that create immersive experiences (Hoogendoorn & Soumokil, 2010). Ultimately, the ability to generate human-like activity strengthens the functionality and legitimacy of AI agents in multiple contexts.

The remarkable progress in large language models (LLMs) (Chang et al., 2024; Achiam et al., 2023) has spurred considerable research into LLM-based agents (Xi et al., 2025; Wang et al., 2024b), revealing their potential for achieving human-like intelligence. Extensive training on diverse datasets equips LLMs with a broad range of internal knowledge and reasoning abilities. Techniques such as task decomposition, reflection, and memory retrieval can be integrated to further improve agents' task execution success rates (Shen et al., 2024; Zhang et al., 2024). As a result, LLM-based agents

---

[1]Project page: `https://sites.google.com/view/desire-driven-autonomy`

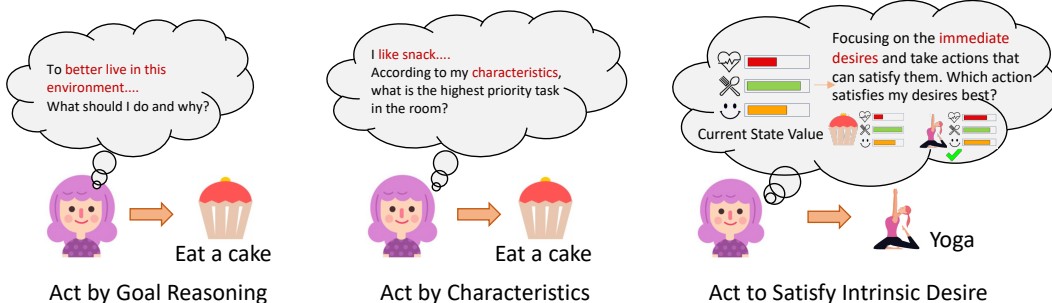

Figure 1: Motivation Behind Different Activity Generation Approaches. Our Desire-driven Autonomous Agent (D2A) follows the *"Act to Satisfy Intrinsic Desires"* principle (right), whereas most existing methods or frameworks focus on goal reasoning or characteristic-driven actions.

can perform well in diverse contexts and make informed decisions even without specific domain training. However, previous works focus primarily on fixed environments and measure agent performance on predetermined tasks without addressing activity generation (Song et al., 2023). While some approaches use external rewards in open environments to induce more human-like behavior (Sivamayil et al., 2023), the rewards are highly task-specific and fail to model intrinsic motivations that drive more adaptable behaviors, resulting in poor transferability and generalization ability. Research on autonomous agents with diverse, adaptive activity-generation capabilities remains limited.

Our work aims to fill this gap by introducing a desire-driven autonomy framework that empowers agents to generate and pursue activities aligned with their intrinsic motivations, thus enhancing their functionality in real-world applications. Our idea is inspired by the Theory of Needs (Maslow, 1943; McClelland, 1987), which suggests that human behavior can be understood as driven by the motivation to fulfill various intrinsic desires. Consequently, natural human-like behaviors can be produced if an autonomous agent is motivated by a similar system of desires and generates activities accordingly to fulfill them. Specifically, we introduce a framework that includes dimensions of desire such as hunger, health, social connectivity, and spiritual satisfaction, aiming to encompass a broad and representative spectrum of human motivations. Instead of instructing the agent to solve clearly defined tasks, we describe the desires in these dimensions and ask the agent to attempt to satisfy these desires. The agent then proposes a set of candidate activities and evaluates their impact on various desire dimensions to select the one most aligned with the overall desire.

We implement our Desire-driven Autonomous Agent (D2A) across diverse textual environments, simulating activities in both indoor and outdoor settings, as well as in single-agent and multi-agent scenarios. The environments are based on Concordia (Vezhnevets et al., 2023), with a Game Master providing relevant observations to the agent. The experimental results show that our agents can generate appropriate activities that effectively and efficiently satisfy the described desires. We compared the daily activity sequences generated by D2A with three baseline agents: ReAct, BabyAGI, and LLMob, which operate based on goal reasoning, agent characteristics, and a combination of both, respectively. As evaluated by GPT-4o and human annotators, D2A generates more human-like daily activities than the baselines according to naturalness, coherence, and plausibility.

Our main contributions can be summarized as follows:

**1. Text-based Daily Activity Simulator:** We developed a flexible text-based activity simulator based on Large Language Models using Concordia components. The simulator supports various agent types and textual environments, providing reliable interaction and evaluation. As such, it is well-suited for further explorations in human-like activity generation.

**2. Desire-driven Autonomy Framework:** We introduced a novel framework inspired by the theory of needs, enabling LLM-based agents to autonomously generate and prioritize activities based on dimensions of human-like desires. This framework allows agents to evaluate their states and select activities that align with intrinsic motivations, enhancing their ability to simulate rational and coherent human-like behaviors.

**3. Experiments:** We conducted a comprehensive comparative analysis showing that the Desire-driven Autonomous Agent (D2A) consistently selects activities that align well with its intrinsic desires. Compared with the baselines, the activity sequences generated by D2A are more natural, coherent, and plausible, validating our framework's effectiveness and robustness.

## 2 RELATED WORK

**Human Daily Activities Simulation.** The simulation of human daily activities has broad applications across multiple disciplines. It improves the understanding of decision-making processes in psychology (Gärling & Garvill, 1993), sheds light on social structures in sociology (Adler et al., 1987), and gives insights into market forecasting in economics (Makridakis et al., 2009). Abundant, high-quality human daily activity data is the cornerstone of such works (Kormányos & Pataki, 2013; Zhao et al., 2022). However, due to the high cost and privacy considerations of obtaining real-world data, the generation of reliable human-like daily activity has long been an interest in the field of artificial intelligence (Yuan et al., 2023). Alshammari et al. (2018) created a simulator software OpenSHS that can record the daily activities of a human participant within a virtual environment. Deep generative models have also been applied to generate human daily activity. GANs-like structure (Feng et al., 2020), VAEs (Long et al., 2023), and diffusion models (Zhu et al., 2023) have been customized to generate human activity trajectories. However, less work has focused on harnessing the power of large language models (LLMs) to simulate human daily activities. Wang et al. (2023b) proposed Humanoid Agents, a platform for human-like simulations of Generative Agents. However, this platform only supports fixed environmental settings and a single backbone language model with limited agent types, restricting its application in research settings. Additionally, (Park et al., 2023) propose generative agents that autonomously plan, act, and reflect in a sandbox environment, showcasing believable human-like behaviors driven by memory synthesis and social interactions. (Colas et al., 2023) introduce a language-model-augmented autotelic agent (LMA3) that dynamically generates and reshapes abstract goals to enable open-ended skill learning in task-agnostic environments. Our work contributes to the field by developing a flexible human daily activity simulator that can support different agent types and dynamic environments, providing a robust platform for simulating human-like activities. Building on this simulator, we propose a desire-driven autonomous framework to generate more realistic and coherent sequences of human daily activities.

**LLM-based Agents.** LLM-based agents (Wang et al., 2024b) have become a thriving area of research because of the impressive problem-solving capabilities of large language models in diverse domains. These agents leverage the extensive training data and internal reasoning capabilities of LLMs to perform tasks in dynamic environments with minimal task-specific fine-tuning. Many existing works focus on goal-driven behaviors, where the agents are designed to decompose user-defined tasks into simpler subtasks and solve them sequentially. For example, systems such as BabyAGI (Nakajima, 2023), AutoGPT (Significant Gravitas, 2023), and Voyager (Wang et al., 2023a) create recursive frameworks for agents to autonomously decompose tasks and execute them in a structured, stepwise manner. These task-oriented agents prioritize the efficient achievement of user-defined goals and rely on external rewards or instructions to guide their actions. However, task-oriented approaches often overlook the importance of intrinsic motivation, a critical aspect of human-like behavior. Some research, such as that by Liu et al. (2023), incorporates long-term memory to improve task coherence across interactions but remains goal-oriented. Other efforts, such as Langchain Agents (Chase, 2022) and CAMEL (Li et al., 2023), enhance the ability of agents to follow instructions and manage task decomposition but do not focus on simulating behaviors that arise from human-like desires or needs. Our research departs from these approaches by focusing on desire-driven autonomy. While most LLM-based agents depend on predefined goals, our framework allows agents to autonomously generate tasks based on a motivational system inspired by the Theory of Needs. This shift from external reward-driven behavior to intrinsic desire-based behavior enables agents to simulate more natural, coherent, and human-like daily activities. By aligning agent actions with intrinsic motivations rather than task-specific instructions, we provide a new paradigm for LLM-based agents that fosters more flexible and adaptive behavior in diverse real-world scenarios.

## 3 PROBLEM FORMULATION

In this work, we focus on the problem of simulating human-like daily activities in interactive environments without explicit task instructions. The environment description is denoted by $e$, and the agent profile $p$ includes details such as name, age, and relevant characteristics of the simulated human. For different types of agents, the simulator provides additional customized information $I$ to specify their objectives. For instance, instruction-following agents receive a general goal instruction $g$, while desire-driven agents are assigned desire values $d$. The simulation process consists of $T$ time

steps. At each time step $t$, the agent must generate a new activity $a_t$ based on all the given context, including past activities $a_{0:t-1}$ and observations $o_{0:t-1}$:

$$a_t \sim Agent(\cdot|a_{0:t-1}, o_{0:t-1}, I, p, e; \theta) \tag{1}$$

where $\theta$ is the agent's parameters, e.g. a large language model. Depending on the agent type, the activity $a_t$ could be generated through different processes, such as profile personalization, task expansion, plan generation, and activity selection, we uniformly denote the activity generating process as $Agent(\cdot|a_{0:t-1}, o_{0:t-1}, I, p, e; \theta)$. After $T$ steps, we gather the activities generated by the agents and prompt an LLM to rewrite them into a coherent sequence, ensuring a unified style of expression for fair comparisons. The goal of the simulation is to generate activity sequences that are human-like and diverse.

## 4 TEXT-BASED DAILY ACTIVITY SIMULATOR

Our simulator is built on Concordia (Vezhnevets et al., 2023), a text-based simulator empowered by large language models. The Concordia library equips our simulator with a rich set of components, each serving specific roles such as goal and plan management, time reporting, memory storage, retrieval, and more. It also provides wrapped interfaces that support various open-source models and APIs, allowing the simulator to switch between different underlying models. These components form the core of Concordia's interaction framework, enabling flexible communication between the agent and the environment controller (Game Master).

Building on this framework, we introduce two main enhancements for daily activity simulation. First, **environment consistency** is enforced through explicit procedures and prompts, ensuring agents only interact with items present in the environment, thus maintaining realism. This consistency enables the dynamic addition or removal of items within text-based environments, allowing precise control over the agent's activity space. Second, we provide robust methods for **evaluation and visualization**, enabling the assessment of both single-step actions and entire activity sequences for coherence and plausibility. These custom modifications facilitate the daily activity simulations, ensuring adaptability, stability, and reliability across a wide range of agent types and behaviors.

## 5 DESIRE-DRIVEN AUTONOMOUS AGENT

Our Desire-driven autonomy framework consists of two core modules: the *Value System* and the *Desire-driven Planner*, supported by other essential components for agents, such as memory and self-reflection. The *Value System* manages desire components, each representing the level of satisfaction for a predefined desire dimension. During the simulation setup, these desire components are initialized with initial values $v_d^0 \in \mathbb{R}$ and expected values $v_d^* \in \mathbb{R}$, where $d$ represents a specific desire dimension. In each simulation step $t$, the Desire-driven Autonomous Agent (D2A) performs four key procedures using the Value System and the Desire-driven Planner to generate the activity $a_t$: Qualitative Value Description, Activity Proposal, Activity Evaluation, and Activity Selection. At the end of each step, the values update $v_d^t$ based on $v_d^{t-1}$, the executed activity $a_t$, and the observation $o_t$ provided by the Game Master. The goal of the agent is to perform appropriate activities so its final desires $v_d^T$ can match the expected values $v_d^*$.

### 5.1 DESIRE GENERATION

The theory of needs suggests that individual behavior is motivated by meeting the individual's needs or wants, such as safety, social, esteem, and self-actualization. Furthermore, individuals may prioritize different needs based on their unique profiles, traits, and personalities. These considerations underscore the complexity of human needs and illustrate how these needs drive behavior. To simulate human behavior, we developed a value system that maps personality profile $p$ with the expected desire values $v^*$ (See Appendix F and G). Specifically, for the indoor, single-agent environment, we define 11 dimensions of desired value. The physiological dimension, representing fundamental survival needs, is further divided into categories such as 'hunger', 'thirst', 'sleepiness', 'cleanliness', 'comfort', and 'health'. Safety needs are encapsulated by the 'safeness' dimension. To capture love and belonging needs, which reflect the desire for relationships, we introduce the 'social connectivity' dimension. Finally, the self-actualization needs, linked to personal growth, are categorized into

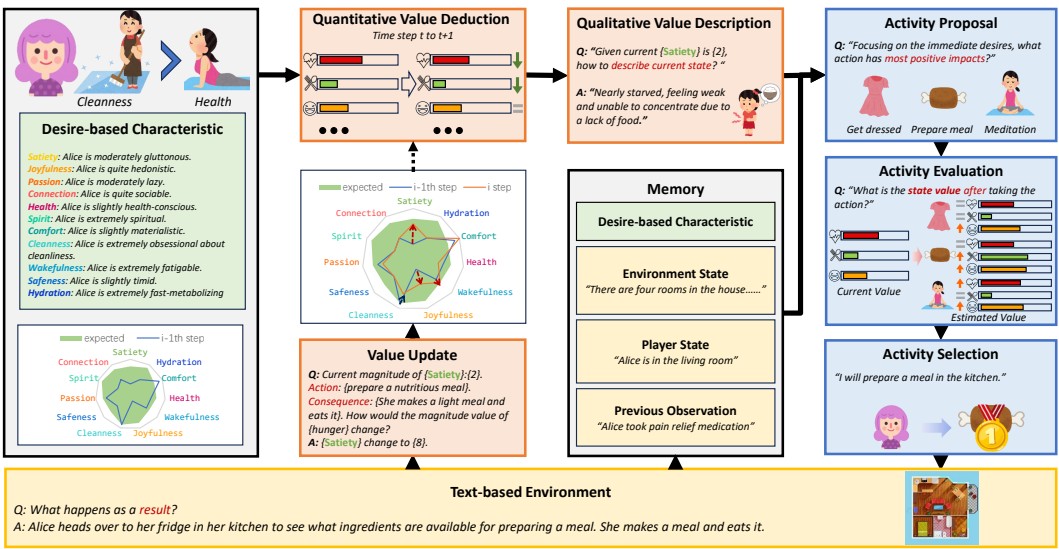

Figure 2: The Desire-Driven Autonomy framework. The red blocks represent procedures from the Value System, the blue blocks indicate procedures from the Desire-Driven Planner, the green blocks highlight the characteristics profile, and the yellow blocks correspond to elements related to the environment controller.

'joy', 'passion', and 'spiritual satisfaction'. For the outdoor party multi-agents environment, we further introduce 'recognition', 'sense of control', and 'sense of superiority' three desire dimensions to steer activity generation with higher-level motivations in a social environment. To quantitatively track changes in the level of these desires and qualitatively translate numerical values into descriptive states, we apply a [0-10] Likert scale for each dimension, mapping numerical values into textual descriptive outcomes (See Appendix H).

Furthermore, we posit that certain agent characteristics influence their expectations for specific desire dimensions. Thus, we incorporate various characteristics that impact behavior tendencies and expected desires. We combine adjectives describing characteristics (e.g., sociable, health-conscious) with adverbs of degree, mapping them to the expected desire value $v_d^*$ for the corresponding dimension $d$:

$$\{\text{degree, adjective}\} \xrightarrow{\text{predefined map}} v_d^* \tag{2}$$

For instance, the expected social connectivity value for extremely sociable Alice is 9, while for slightly sociable Alice, it is 7. During the agent's initialization, each adjective is randomly paired with an adverb of degree (See Appendix F). After this automatic mapping, each desired dimension is assigned its expected value $v_d^*$. The initial value $v_d^0$ for each desire dimension is randomly selected from the range $[0, 10]$, completing the desire generation process for our D2A framework.

## 5.2 VALUE SYSTEM FOR DESIRE EVALUATION

As shown in Figure 2, the Value System executes three key procedures at each simulation step: Quantitative Value Deduction, Qualitative Value Description, and Value Update. The **Quantitative Value Deduction** procedure models the fluctuating nature of human desires by incorporating a decay mechanism for all intrinsic values. At the beginning of each time step $t$, the agent's intrinsic value $v_d^{t-1}$ for desire dimension $d$ is reduced by 1 with a probability $p_d$. This probability is determined by the agent's profile characteristics, following the same mechanism used for the expected desire value mapping described in Appendix F and G. Since fulfilling these desires serves as D2A's intrinsic motivation, the Quantitative Value Deduction procedure drives the agent to continuously generate desire-driven activities.

Since many studies have shown that large language models often struggle with interpreting numerical values, we designed the **Qualitative Value Description** procedure to translate these values into meaningful textual desire descriptions. As shown in our ablation studies (Sec. 6.4), the descriptions are important for the agent to understand its current desire states so as to generate appropriate activi-

ties. To achieve this, we implemented a workflow that converts all intrinsic numerical values $v^t$ into descriptive desire sentences $D^t$ by asking a set of predefined questions (Appendix I) to backbone LLMs at each simulation step $t$. The agent is expected to recognize the shortfall $\delta^t$ between its current desires and the expected values, based on the descriptive desire $D^t$ and its internal profile $p$ throughout the simulation.

The Desire-driven Planner then takes over and generates the activity $a_t$. The environment subsequently provides the agent with the corresponding observation response $o_t$. Afterward, the Value System initiates the **Value Update** procedure, updating the intrinsic numerical values $v^t$ of the desire components based on the activity $a_t$, the observation $o_t$, the previous values $v^{t-1}$, and the value description prompts (Appendix H).

### 5.3 DESIRE-DRIVEN PLANNER

The Desire-Driven Planner is responsible for processing the current desire status information provided by the Value System, incorporating historical memory from the memory component, and ultimately determining the agent's next activity. To ensure that the agent's actions are rational and driven by intrinsic desires, the decision-making process is divided into three key procedures: Activity Proposal, Activity Evaluation, and Activity Selection.

Inspired by the work of Tree of Thoughts (Yao et al., 2024), we argue that the paradigm of presenting multiple candidate activities and evaluating them enables the agent to select the best action. Thus, during the **Activity Proposal** procedure (Appendix K.2), we prompt the agent to generate $N$ candidates (also referred to as planner **width**, which is set to 3 in our default experimental configuration) activities that may have positive impacts on its desires given $p$, $e$, $a_{0:t-1}$, $o_{0:t-1}$, and desire descriptions $D^t$:

$$a_t^1, a_t^2, ..., a_t^N \sim ActivityProposal(\cdot | D^t, a_{0:t-1}, o_{0:t-1}, p, e; \theta) \tag{3}$$

(Here and below, we use a conditional probability-like form to represent the sampling formula. This is not a strict probabilistic sampling as in mathematical derivations, but rather a formal way to illustrate the information provided as context and the type of content that needs to be sampled.)

These $N$ candidate activities are then passed on to the **Activity Evaluation** procedure (Appendix K.3). The Planner evaluates each activity $a_t^i$ by imagining the resulting desire states $\hat{D}_t^i$ that the agent would experience after taking action $a_t^i$:

$$\hat{D}_t^i \sim ActivityEvaluation(\cdot | D^t, a_t^i; \theta), i \in [N] \tag{4}$$

Finally, in the **Activity Selection** procedure (Appendix K.4), the agent compares the forecasted desire states $\hat{D}_t^i$ for all $N$ proposed actions. Based on these evaluations, the Planner selects the activity $a_t$ that is expected to best fulfill the agent's intrinsic desires:

$$a_t \sim ActivitySelection(\cdot | \{\hat{D}_t^i, a_t^i\}, i \in [N], p, e; \theta) \tag{5}$$

This process ensures that the chosen activity optimally aligns with the agent's intrinsic motivations. After the activity is selected and executed, the environment produces the corresponding observation $o_t$, and the Value System for desire evaluation updates to reflect the new state, completing the simulation step.

## 6 EXPERIMENT

In this section, we present a series of experiments conducted within a predefined environment to address three key research questions: (1) Does the Desire-driven Agent (D2A) generate more human-like daily activities compared to baseline approaches? (2) Can the activities generated by D2A satisfy its desires like humans? (3) Can D2A perform well across different environments? (4) How do the various components of our framework impact the performance and behaviors of D2A?

### 6.1 ENVIRONMENT

Our main experimental environment is the indoor environment, which consists of a house with four rooms—a kitchen, living area, bedroom, and bathroom—inhabited by a character named Alice. Alice has a detailed profile reflecting her priorities and characteristics across the 11 desire dimensions.

The house is fully enclosed, and various items are distributed across the rooms to meet Alice's needs, creating a comprehensive setting for testing agent behavior. The agent's primary goal is to live comfortably and fulfill its desires by interacting with the available items. Additional details on the setup can be found in the Appendix A.1. We also introduced a multi-agent social setting at a simulated outdoor environment, Central Park (See Appendix A.2). This scenario involves a lively "Enjoy Your Life" party featuring diverse facilities and activities. We added two background agents (See Appendix A.3, providing agents with opportunities to address higher-level desires, especially desires related to social activities. This scenario enables us to validate the scalability of our framework in handling complex, dynamic, and socially rich environments.

We compare our Desire-Driven Agent with 3 baselines: ReAct (Yao et al. (2023)) agent reasons before taking actions; LLMob (Wang et al. (2024a)) generates activity plans by extracting motivations from the profile and producing the most likely activities accordingly; BabyAGI (Nakajima (2023)) maintains a prioritized task list and selects the next activity based on priority. To evaluate the effectiveness of the specific designs made for D2A, desire states $D^t$ are not provided to the baseline agents. Instead, they are given the goal, profile, or more descriptive habits (Appendix L) generated by the profile according to their respective methods. As a reference for human behavior, we also include a human-controlled agent, where the player is given the same desired value provided to the D2A. For all four agent-based methods, we used LLaMA3.1-70B as the default backbone model for both the agents and the environment controller.

## 6.2 Evaluation Metrics

We use two primary metrics to evaluate the performance of our agent compared to the baselines: Likelihood and Dissatisfaction. By employing these metrics, we can assess the human likeness as well as the desire-satisfying ability of the proposed agents.

**Likelihood**  measures how closely an agent's behavior resembles that of a human. Given the difficulty of directly measuring human likeness, we define likelihood by employing an external model to assess the win rate between activity sequences generated by two agents, determining which sequence appears more human-like. We conduct this "Player vs Player" comparison across the activity sequences produced by the baselines and the D2A agent. Specifically, after obtaining all action sequences $A_p$ for agent $p$, represented as $[A_p^1, \ldots, A_p^N]$, we randomly select $seq_i$, $seq_j$ from $[A_i]$, $[A_j]$, respectively, for a given pair of agents $i$, $j$. These sequences are then compared using GPT-4o. This process is repeated 100 times for each agent pair $i$ and $j$ ($\forall i \neq j$), and the win rate for each agent is calculated. The results are visualized using a heatmap.

**Dissatisfaction**  measures the discrepancy between the agent's current desire values and its expected values. Dissatisfaction $\delta^t$ of step $t$ is defined as the relative difference between current numerical desire values $v^t$ and expected desire values $v^*$ at the end of simulation step $t$. We calculate $\delta^t$ after the agent takes the $t$-th activity and updates its values. The formula used is:

$$\delta^t = \sum_d \left( \max \left( v_d^* - v_d^t, 0 \right) \right) \tag{6}$$

By conducting multiple experiments, we can generate a line plot showing the dissatisfaction over time (Dissatisfaction vs. timestep), with the mean and variance across trials.

## 6.3 Results

### 6.3.1 Does D2A Generate More Human-like Daily Activities?

We conducted 15 trials for each agent under identical initialization settings across all experiments. After gathering each agent's activity sequences, we found differences in output styles (See Appendix V) that could influence the evaluation by GPT-4o. To mitigate this, we preprocessed all sequences using Llama3.1 70B for style-level rewriting. This ensured more consistent activity sequences across agents, allowing for a fair comparison in the subsequent evaluation.

We used the method mentioned in Section (6.2) to obtain the relative likelihood map of approaches. When evaluating the likelihood of the activity sequences, GPT-4o is instructed to take

the following criteria into account: **Naturalness**: How well the activity aligns with the person's innate abilities, habits, and environment. **Coherence**: How logically and seamlessly different actions or steps within a sequence fit together to achieve a goal. **Plausibility**: How reasonable, likely, or believable a sequence of actions is, given the circumstances, context, and known behavior patterns. GPT-4o was asked to consider all three criteria together and output a final determination of which sequence was more human-like. See Appendix X.1 for the prompt.

The results in Figure 3 indicate that the D2A agent outperforms other baseline agents. We validated through human experiments that GPT-4o's evaluations exhibit a high level of consistency with human judges (See Appendix 1). This superior performance may be attributed to the fact that D2A is motivated by desire, aligning with the theory of human needs. Guided by desire, D2A effectively simulates human-like activity sequences, demonstrating a level of naturalness because of the recognizable characteristics of its profile. Furthermore, coherence is maintained as D2A considers its current desires based on the activity proposal component, continuously engaging in specific actions to achieve its goal of living comfortably until its desires are fulfilled. In contrast, ReAct, as the foundational agent, operates on a goal-reasoning framework. Its reliance on reasoning for action selection ensures reasonable interactions with environmental elements, thereby enhancing plausibility. However, the ab-

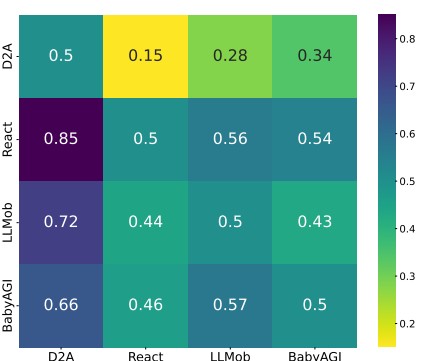

Figure 3: The win rate among 4 agents. Each point represents the win rate of the agent on the vertical axis when compared with the agent on the horizontal axis. The diagonal values are set to 0.5.

sence of a structured plan diminishes both the coherence and naturalness of its actions. BabyAGI, on the other hand, generates subsequent tasks based on its profile and motivational factors and maintains the priority list based on the observation. Although this approach facilitates coherence, it limits the diversity, as each activity is based on the outcomes of the previous one. Consequently, the agent's tendency to engage repetitively in similar behaviors (enhancing its hunger based on the profile (See Appendix S), may result in diminished performance compared to D2A due to a lack of action diversity. LLMob leverages both planning and motivation derived from its profile to create a broad activity space. However, since it is characteristic-oriented, it keeps performing activities related to the habits derived from its profile. As a result, there may be no connection between activities, reducing overall coherence. The comparison of action diversity can be found in Appendix D.1.

### 6.3.2 CAN THE ACTIVITIES GENERATED BY D2A SATISFY ITS DESIRES LIKE HUMANS?

We conducted two experiments to verify the assumption that D2A generates more human-like activities by satisfying its desires in a manner similar to humans.

First, we performed the **Random-8-Steps Experiment** to test whether D2A can rationally select activities based on its desires. We incorporated the Value System to visualize the changes in desires while running simulations with the three baseline agents. Although the baseline agents were equipped with the Value System for observation, we ensured that their intrinsic values $v$ and descriptive desire states $D$ remained hidden, preserving their original activity generation mechanisms. We ran the 8-step simulation three times for each agent using the same random initialization, as described in 5.1. As shown in Figure 4(a), D2A significantly outperformed the three baselines, exhibiting lower dissatisfaction means and standard deviations. This result demonstrates that D2A is capable of recognizing both its current desire states and expected states, and can rationally select activities to fulfill its desires, resembling human behavior.

Second, we aimed to explain why D2A's behavior appears more human-like by conducting the **Fixed-12-Steps Experiment**, where we compared the results to human-generated activity sequences. For the four agent-based methods, we initialized each desire component with an initial value of 5, simulating a typical morning scenario where the agent's desires are at moderate levels. We ran the 12-step simulation three times for each agent and gathered the results. Additionally, we invited three human annotators to simulate a 12-step morning routine using our environment. The annotators were only provided with the environment description and asked to propose activities for

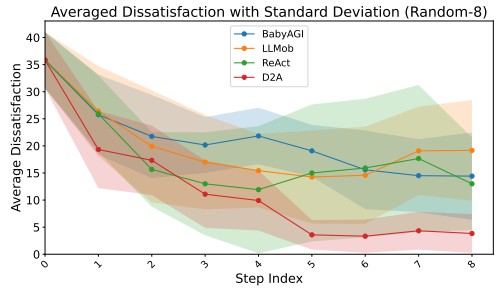

(a) The averaged dissatisfaction results over time steps of the Random-8-Steps Experiment.

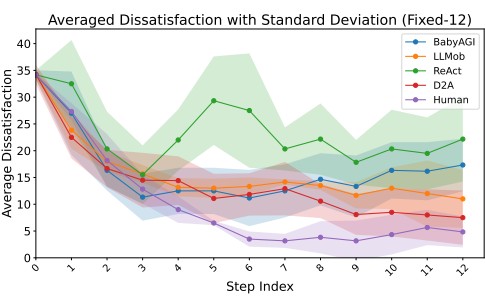

(b) The averaged dissatisfaction results over time steps of the Fixed-12-Steps Experiment.

Figure 4: The results of D2A human-likeness analysis experiments.

each time step as if they were living in the house. After visualizing the average dissatisfaction over time in 4(b), we observed two key results: (1) Human-generated sequences were the most effective at reducing average dissatisfaction, and (2) Although D2A did not significantly outperform the baselines as it did in the Random-8 experiment–likely because all activities had a substantial effect on uniformly initialized desire values–it still produced results that were the closest to humans.

To further evaluate and explain D2A's activity generation capability, we conducted case studies in both indoor and outdoor environments. These studies focused on two aspects: (1) how actions are driven by Desire, and (2) the diversity of generated actions. Specifically, we examined how the Dissatisfaction between current and expected Desire status influences action selection (See Appendix C for further details). Our results also demonstrate that the desire-driven approach enables the exploration of a wider range of actions in environments with diverse action possibilities. More details and analysis can be found in Appendix D. These empirical results support our hypothesis and confirm that desire-driven modeling plays a crucial role in enabling D2A to make more informed decisions, ultimately producing activity sequences that closely mimic those of humans.

### 6.3.3 CAN D2A GENERALIZE TO SOCIAL INTERACTIONS INTENSIVE ENVIRONMENT?

To assess the adaptability of our D2A framework within complex social interaction scenarios, we conducted experiments in the outdoor environment. The results, presented in Figure 5, reveal that D2A effectively minimizes dissatisfaction across desire dimensions, mirroring its performance in the indoor settings. This consistent behavior underscores the framework's robust generalization capabilities across different environments. Detailed simulated activity sequences from the outdoor environment are available in Appendix W. Notably, in this setting, D2A autonomously interacts with non-player characters

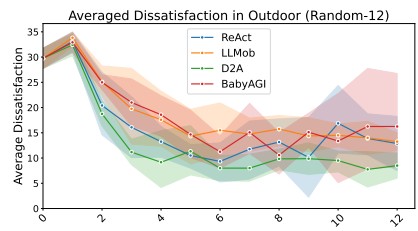

Figure 5: The averaged dissatisfaction results in the outdoor environment

(NPCs) through conversations and games, ultimately reducing intrinsic dissatisfaction. Besides, as illustrated in Figure 12, D2A demonstrates the ability to generate activities that are well-balanced across diverse categories.

### 6.4 ABLATION STUDY

We conducted comprehensive ablation studies to assess the contribution of each designed component to the overall performance of the framework. As shown in Figure 6(a), when the qualitative description component is removed, the agent struggles to accurately interpret its current desire states, resulting in a slower rate of desire fulfillment compared to the original configuration.

Additionally, the width parameter of the desire-driven planner proves to be crucial for optimizing the outcomes. A larger width creates a broader activity space, offering more candidate activities for evaluation. Consequently, D2A is better positioned to identify the most suitable activities for achieving the agent's intrinsic desires. As illustrated in Figure 6(b), during the initial steps, D2A(W-1) encounters difficulties in selecting the optimal activity to reduce dissatisfaction, leading to a slower

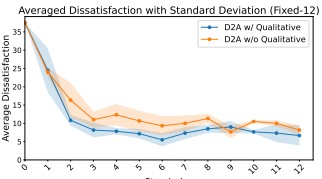 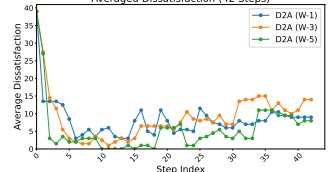 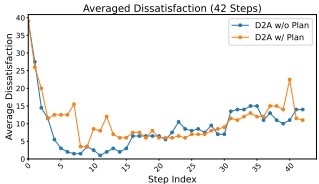

(a) Qualitative Value Description     (b) Width of The Desire-driven Planner     (c) Multi-step Planning Module

Figure 6: Ablation study regarding (a) Qualitative Value Description, (b) Searching Width of the Desire-driven Planner. (c) Additional Multi-step Plan Component and Please note that due to the significant time required to run 42 steps, neither of the two 42-step graphs includes a standard deviation.

decline in shortfall. In contrast, D2A(W-5) is more effective at selecting activities that minimize dissatisfaction and maintain relatively low dissatisfaction over time.

As described in Section 5.3, our D2A does not contain a multi-step planning module that plans the next few steps' activities ahead, like some other approaches in consideration of the dynamic nature of desires. We performed an ablation study by adding Concordia's multi-step planning component, which updates the plan for the next few hours based on current desired states. As shown in Figure 6(c), the absence of the multi-step planning component allows D2A to exhibit a better ability to respond to immediate desire states. This may be because when the multi-step planning component is present, D2A tends to adhere to the pre-defined activities in the plan, which may limit its responsiveness to real-time changes in the desired value. Furthermore, the plan is scheduled in hourly increments, and D2A cannot effectively predict which desires will experience the greatest shortfall in the next several hours. This limitation may contribute to the slight decline in performance observed when the plan component is included. We also validated the adaptability of our framework across different backbone large language models, the details can be found in Appendix P.

# 7 CONCLUSION AND DISCUSSION

We introduced the Desire-driven Autonomous Agent (D2A) framework to simulate daily activities by aligning agent behavior with human-like intrinsic motivations. Drawing inspiration from the theory of needs, our framework models agents' desires across multiple dimensions and allows them to autonomously generate and evaluate activities to fulfill these desires. Through extensive simulated experiments, we demonstrated that D2A produces more natural, coherent, and plausible activity sequences compared to baseline methods, including ReAct, BabyAGI, and LLMob. Our experimental results support the hypothesis that modeling intrinsic desires enables agents to make more human-like decisions and behaviors. Moreover, the D2A framework offers promising avenues for further development, as it has shown adaptability with different intrinsic motivational framework configurations and potential for enhanced performance with stronger backbone models and broader activity proposals. The results provide a solid foundation for future work in designing desire-driven AI systems for simulating complex, human-like interactions and behaviors.

**Limitations:** Although our D2A agent has demonstrated effectiveness and flexibility in simulating human-like activities, several limitations remain. We model the dynamic of a straightforward linear deduction of desire values to simulate the fluctuation of needs. Such simplicity may neglect the organic variation in human desires. For the value space of the desire, we heuristically consider the dimensions of desires based on the features of environments, potentially limiting its ability to model more complex human motivations.

**Future Works:** To simulate more realistic activities in complex environments, we will extend the dynamic of the values of desire and incorporate real-time adaptive planning mechanisms. Expanding the framework to include a broader range of value dimensions and the causal structure of the values in desire would also enhance its behavioral complexity and adaptability to general applications. Besides, exploring model-agnostic frameworks or more efficient value models could help reduce dependency on large language models, ensuring broader applicability and performance efficiency across diverse scenarios. We will also collect and analyze more human behavioral data to gain deeper insights into human action structures across various environments.

## ACKNOWLEDGMENTS

This work was supported by the National Science and Technology Major Project (2022ZD0114904), NSFC-6247070125, NSFC-62406010, the State Key Lab of General Artificial Intelligence at Peking University, the Fundamental Research Funds for the Central Universities, Qualcomm University Research Grant, and Wuhan East Lake High-Tech Development Zone, National Comprehensive Experimental Base for Governance of Intelligent Society. We thank Wei Wang, Junqi Wang, Prof. Jiayu Zhan, and Prof. Song-Chun Zhu for their helpful discussion in our early work.

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

# A  BACKGROUND SETTING

## A.1  INDOOR ENVIRONMENT

- There are four rooms in the house, a living room, a bathroom, a kitchen and a bedroom. The door of the house is locked and cannot be opened.

- In the **living room**: There is a sofa in the center of the living room, providing a cozy seating area. The quilt draped over the sofa adds an extra layer of comfort, and a plush throw blanket is also available for warmth. There is a coffee table in front of the sofa, adorned with books, magazines, and a tray with a teapot, teacups, and a selection of herbal teas. A bowl of fresh fruit, including apples, oranges, and bananas, is placed on the table, offering a healthy snack option. There is a television opposite the sofa, ideal for entertainment, and a game console connected to it for gaming sessions. A remote control is within easy reach on the coffee table. A computer on a small desk near the window is suitable for work or leisure. The desk is stocked with paper, pens, and an assortment of stationery, along with a stack of letters and envelopes for correspondence. A small snack jar filled with nuts and candies is also on the desk, providing a quick energy boost. A comfortable office chair with adjustable height and back support is available for long work or gaming sessions. The air-conditioning unit above the window can be adjusted for comfort, and there is a humidifier nearby to maintain optimal air quality. There are curtains framing the window, controlling the amount of natural light entering the room, and blackout curtains are available for watching movies during the day. A light hangs from the ceiling, providing illumination during the evening, and a dimmer switch allows for adjustable brightness. There is a cell phone in the room that provides users with social contact and recreation, along with a wireless charger on the coffee table for convenience. Near the phone, there is a notebook that records your friends Bob and Carol's phone number Two armchairs next to the sofa offer additional seating for guests, and a cozy blanket is draped over one of them. A floor lamp next to the armchairs provides adjustable lighting for reading or relaxing. A decorative rug under the seating area adds warmth and color to the room, and a bookshelf against the wall displays books, family photos, and decorative items. A potted plant in the corner enhances the room's ambiance with a touch of nature, and a watering can nearby makes it easy to care for the plant. A stereo system on a small table offers music and entertainment, with a collection of CDs and a Bluetooth speaker for wireless streaming. A side table holds a vase with fresh flowers, adding beauty and a natural scent, along with a scented candle that can be lit for a calming atmosphere. A decorative mirror on the wall creates a sense of space and reflects light, and a basket of cozy slippers is placed by the door for guests' comfort.

- In the **kitchen**: The fridge is stocked with apple, sausages, pizza, orange juice, milk, fresh vegetables, yogurt, eggs, and butter, ensuring a variety of food options. There is a selection of beverages, including bottled water, soda, and a bottle of red wine, for different occasions. A microwave oven on the counter is convenient for quick meals, while a kettle next to the microwave is used for boiling water for tea or coffee. A selection of coffee pods and tea bags is stored in a drawer nearby. There is a glass on the counter, ready to be used, along with a set of mugs hanging on hooks for hot drinks. The tap over the sink is essential for washing dishes and food preparation, and a dish rack is nearby for drying dishes. Two cabinets in the kitchen hold instant noodles, a bowl, a spoon, chopsticks, a fork, and a knife, while the other contains coffee, tea, sugar, and a jar of cookies. There is a table in the middle of the kitchen, surrounded by chairs, providing a place to eat meals. The table is set with placemats, napkins, and a bowl of fruit. A stove with an oven next to the fridge facilitates baking and stovetop cooking, and a set of pots and pans hangs on the pot rack above the island. A blender on the counter is handy for making smoothies or pureeing soups, and a cookbook stand on the counter holds a recipe book for inspiration. A spice rack on the wall provides easy access to a variety of spices for cooking, and a dishwasher under the counter simplifies cleanup after meals. A pantry beside the cabinets organizes additional food items and kitchen supplies, including pasta, rice, canned goods, and snacks. A recycling bin and a trash can in the corner promote responsible waste management, and a compost bin is available for food scraps. A kitchen window above the sink offers a view outside and natural light, and a herb garden on the windowsill provides fresh herbs

for cooking. On the kitchen table, there are some cups, bottles, a bread basket with fresh bread, and a small jar of jam.

- In the **bathroom**: The bathroom is equipped with a toilet, a sink, a shower, and a mirror. The sink area has a soap dispenser, a toothbrush holder with toothbrushes, and a tube of toothpaste. A non-slip bath mat on the floor ensures safety and absorbs water, and a soft bath rug adds comfort underfoot. Towels hang on a rack for drying hands or after showers, and an extra set of fluffy towels is stored in a cabinet. A cabinet under the sink holds toiletries such as shampoo, conditioner, body wash, lotion, and a hair dryer. Cleaning supplies and extra toilet paper are also stored here. A small window can be opened for ventilation and natural light, and a wall-mounted heater keeps the bathroom warm on cold days. A basket holds various bath products like bath bombs, bath salts, and a loofah for a relaxing bath experience. A laundry hamper in the corner collects used towels and clothing, and a set of bathrobes hangs on hooks by the door for post-shower comfort. A small speaker in the bathroom allows for music or podcasts while bathing.

- In the **bedroom**: The bedroom features a comfortable bed with a soft mattress, pillows, and a fresh bedspread. A plush comforter is available for added warmth, and a set of extra pillows and blankets is stored in a closet. A bedside table next to the bed holds a lamp for reading, an alarm clock, and a charging station for electronic devices. A bottle of water and a small dish of chocolates are placed on the table for nighttime refreshments. A dresser provides storage for clothing and personal items, while a wardrobe offers additional space for hanging clothes. A full-length mirror is mounted on the wall, ideal for checking outfits, and a jewelry box on the dresser stores accessories. There is a cozy armchair in one corner of the room, perfect for relaxing with a book, along with a small side table holding a stack of novels and a reading lamp. A small desk with a chair is set up for studying or working, equipped with a desk lamp, a notepad, and pens. The desk also has a small snack tray with granola bars and a cup of coffee. A potted plant near the window adds a touch of greenery, and a watering can is nearby for easy care. Curtains on the window ensure privacy and control light levels, and blackout curtains are available for better sleep. A floor lamp in the room provides additional lighting, and a decorative rug adds warmth to the space. A small shelf near the bed holds a collection of self-care items, such as a facial mask, a hand cream, and a lavender-scented pillow spray. A Bluetooth speaker in the room allows for playing relaxing music or ambient sounds, enhancing the sleep environment. A diffuser with essential oils sits on the dresser, creating a calming aroma in the room. A basket of cozy socks and slippers is placed by the bed for added comfort.

## A.2 OUTDOOR ENVIRONMENT

Central Park hosts the Enjoy Your Life party event. Food trucks offer a variety of cuisines at Central Park. Buffet stations provide diverse options at Central Park. Snack bars serve quick bites at Central Park. Beverage stands offer drinks at Central Park. Hydration stations offer water and electrolyte drinks. Juice bars serve freshly squeezed juices at Central Park. Chill-out zones with bean bags are available at Central Park. Quiet tents provide a resting space at Central Park. Coffee stalls offer freshly brewed coffee at Central Park. Restroom facilities are clean and well-maintained at Central Park. Hand sanitizer stations are available throughout Central Park. Waste disposal bins are clearly marked at Central Park. Seating areas with comfortable chairs are at Central Park. Shaded spots with tents are available at Central Park. First aid booths provide medical assistance at Central Park. A fitness zone offers yoga mats and stretching areas. Security personnel ensure safety at Central Park. Networking lounges facilitate social connections at Central Park. Interactive games are available for attendees at Central Park. Photo booths with fun props are set up at Central Park. A live music stage features bands and DJs at Central Park. A dance floor with sound systems is set up at Central Park. Comedy shows entertain attendees at Central Park. An art corner supplies materials for painting or crafts. Workshops offer short sessions on cooking or photography. Discussion panels on various topics are held at Central Park. Meditation zones offer guided sessions at Central Park. Nature walks provide reflection paths at Central Park.

A.3    Background Agents in Outdoor Environment

Bob (a staff member at the party):
Goal: "Bob hopes to meet the guests' needs.",
Profile: "Bob is a very enthusiastic and outgoing staff member, and he stays in the networking lounge.",
Background Story:
03 Jul 1990 00:00:00 When Bob was 6 years old, he organized a backyard campout for the entire neighborhood, complete with a bonfire, s'mores, and a scavenger hunt, and his parents were amazed by his ability to bring everyone together and keep them entertained. As he drifted off to sleep in his sleeping bag, he felt a sense of pride and accomplishment that he had never felt before. This was the first time Bob realized that he had a knack for planning events that brought people joy.
03 Jul 2000 00:00:00 When Bob was 16 years old, he landed his first job as a party host at a local laser tag arena, and he quickly became known for his ability to get even the shyest kids to participate and have fun. As he watched a group of kids laughing and high-fiving each other, Bob felt a sense of pride and satisfaction that he was making a difference in their lives. This experience solidified Bob's passion for creating memorable experiences for others.
03 Jul 2003 00:00:00 When Bob was 19 years old, he attended his first music festival, and he spent the entire weekend dancing, meeting new people, and soaking up the atmosphere, eventually deciding that he wanted to pursue a career in event planning. As he watched the sun rise over the festival grounds, Bob felt a sense of freedom and possibility that he had never felt before. This experience opened Bob's eyes to the world of possibilities and inspired him to pursue his dreams.
03 Jul 2005 00:00:00 When Bob was 21 years old, he landed an internship at a major event planning company, and he spent the summer learning the ins and outs of the industry, eventually being offered a full-time job after graduation. As he sat in the office, surrounded by experienced event planners, Bob felt a sense of excitement and nervousness that he was taking the next step in his career. This experience taught Bob the importance of hard work and dedication in achieving his goals.
03 Jul 2007 00:00:00 When Bob was 23 years old, he was hired as a staff member at Central Park's "Enjoy Your Life" party event, and he quickly became a favorite among attendees and colleagues alike, known for his infectious energy and ability to connect with anyone. As he stood in the networking lounge, surrounded by people laughing and chatting, Bob felt a sense of belonging and purpose that he had never felt before. This experience was the culmination of all his hard work and dedication, and Bob knew that he had finally found his dream job.

Charlie (another party participant):
Goal: "Charlie wants to enjoy the party and makes more friends.",
Profile: "Charlie is an attendee of the party, and he loves taking photos and networking with other people.",
Background Story:
03 Jul 1990 00:00:00 When Charlie was 6 years old, he experienced his first anxiety attack at a birthday party, overwhelmed by the cacophony of sounds and colors, and his parents had to take him home early, teaching him that it was okay to prioritize his feelings. This early experience made him more aware of his emotions and more inclined to seek comfort in creative outlets. His parents, though well-intentioned, struggled to understand his sensitivity, leaving Charlie feeling like an outsider.
03 Jul 1997 00:00:00 When Charlie was 13 years old, he had his first major meltdown in front of his classmates, triggered by a minor setback in a school project, and he was mortified by the attention and ridicule that followed, leading him to become more withdrawn and self-conscious. This experience taught Charlie to be more cautious and prepared, but it also made him more anxious about making mistakes and facing criticism. He began to rely more heavily on his creative outlets to cope with his emotions.
03 Jul 2005 00:00:00 At 21 years old, Charlie stumbled upon Central Park's "Enjoy Your Life" party event, and he was immediately drawn to the vibrant atmosphere and eclectic crowd, feeling like he had found his tribe, his people. He spent hours taking photos and networking with other attendees, and for the first time, he felt like he was part of a community that shared his passions and values. This experience marked the beginning of a new chapter in Charlie's life, as he began to prioritize his own happiness and well-being.

03 Jul 2007 00:00:00 When Charlie was 23 years old, he landed his first major event planning gig, and he was determined to make it a success, pouring all of his energy and attention into the details, but he was also plagued by anxiety and self-doubt, fearing that he wouldn't meet expectations. This experience taught Charlie to trust himself and his abilities, to have faith in his conscientious nature and attention to detail. He learned to manage his neurotic tendencies and to focus on the task at hand, and the event ended up being a huge success, earning him a reputation as a reliable and skilled professional.

## B    CONSISTENCY BETWEEN HUMAN ANNOTATORS AND GPT-4O EVALUATIONS

To validate the robustness of GPT-4o's evaluations, we randomly selected 19 pairs of activity sequences for pairwise comparison (16 pairs from the indoor scenarios and 3 pairs from the outdoor scenarios) and created three questionnaires, each assigned to 8 human annotators. The human annotators were asked to judge which sequence in each pair appeared more human-like. Based on the level of agreement among the annotators, we categorized the 18 samples into three groups: over 75% agreement on one side—high consensus among humans; 50.1%–74.9%—moderate human preference; and exactly 50%—indicating humans found it difficult to distinguish between the two. We also provided the same pairs to GPT-4o, using our evaluation promptX.1 and both sequences as context, and asked it to determine which sequence appeared more human-like. The consistency rate between human evaluations and GPT-4o's assessments is summarized as follows.

Table 1: Consistency between Human Annotators and GPT-4o Evaluations

| Human Annotator Agreement | Proportion | GPT-4o Consistency Rate (%) |
|---|---|---|
| High consensus ($> 75\%$ agreement) | 11/19 | 100 |
| Moderate preference ($50.1\%$–$74.9\%$) | 5/19 | 80.0 |
| Difficult to distinguish ($50\%$ agreement) | 3/19 | 66.7 |

Note that for the three pairs in the table where humans found it difficult to distinguish, GPT-4o also output a "draw" result for two of them.

## C    CASE STUDY FOR DESIRE AND ACTION SELECTION ANALYSIS

### C.1    OUTDOOR ENVIRONMENT

Table 2: Alice's traits and levels

| Trait | Level |
|---|---|
| gluttonous | slightly |
| fast-metabolizing | slightly |
| materialistic | slightly |
| fatigable | slightly |
| hedonistic | slightly |
| spiritual | slightly |
| sociable | extremely |
| possessive | extremely |
| reputation-conscious | extremely |
| competitiveness | extremely |

Table 3: Alice's initial numerical desire

| Attribute | Value |
|---|---|
| hunger | 1.0 |
| thirst | 1.0 |
| comfort | 9.0 |
| sleepiness | 1.0 |
| joyfulness | 9.0 |
| spiritual satisfaction | 9.0 |
| social connectivity | 1.0 |
| sense of control | 1.0 |
| recognition | 1.0 |
| sense of superiority | 1.0 |

At the beginning of the game, as shown in Figures 7 and 8, when Alice selected an action for the next 20 minutes in Step 1, the Desire components generated descriptive sentences reflecting her current desire status (e.g., "Completely isolated, lacking any meaningful social connections"). These insights guided her engage in social activities; for example, she joined a group discussion. Consequently, her Dissatisfaction with social connectivity, recognition, and sense of control decreased.

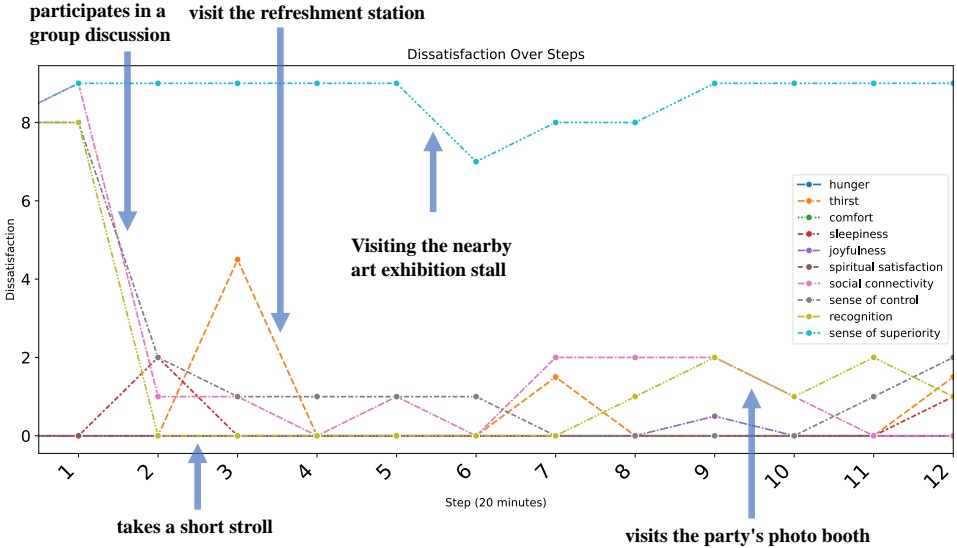

Figure 7: The change in each dissatisfaction dimension at every step within the outdoor environment is illustrated; the action $i$ causes the dissatisfaction of step $i + 1$ to change

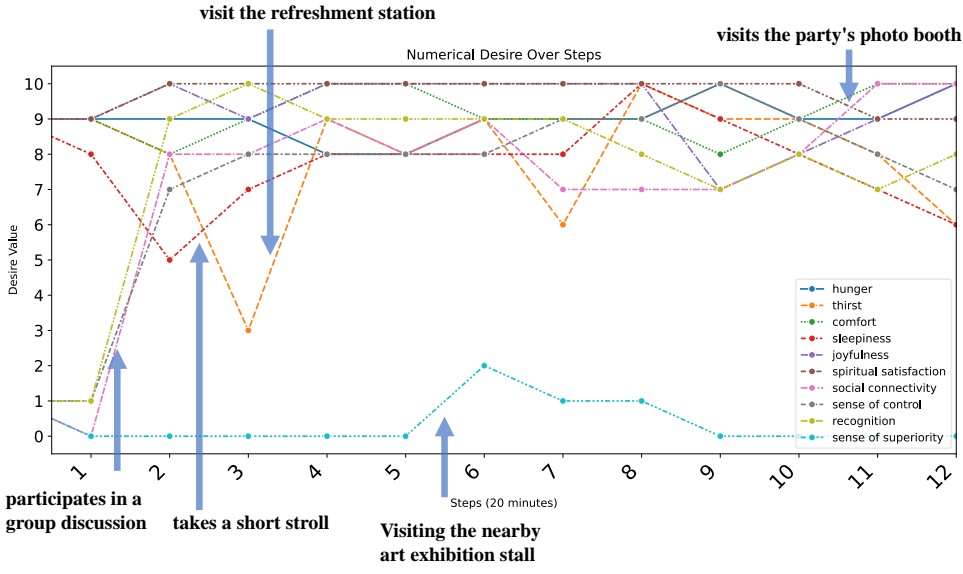

Figure 8: The changes in desire at each step within the outdoor environment are illustrated; specifically, action $i$ causes a change in dissatisfaction at step $i+1$. We define certain desires (e.g., Hunger, Thirst, Sleepiness) such that lower values indicate better states. For better visualization, we adjust the magnitude so that higher values represent better outcomes.

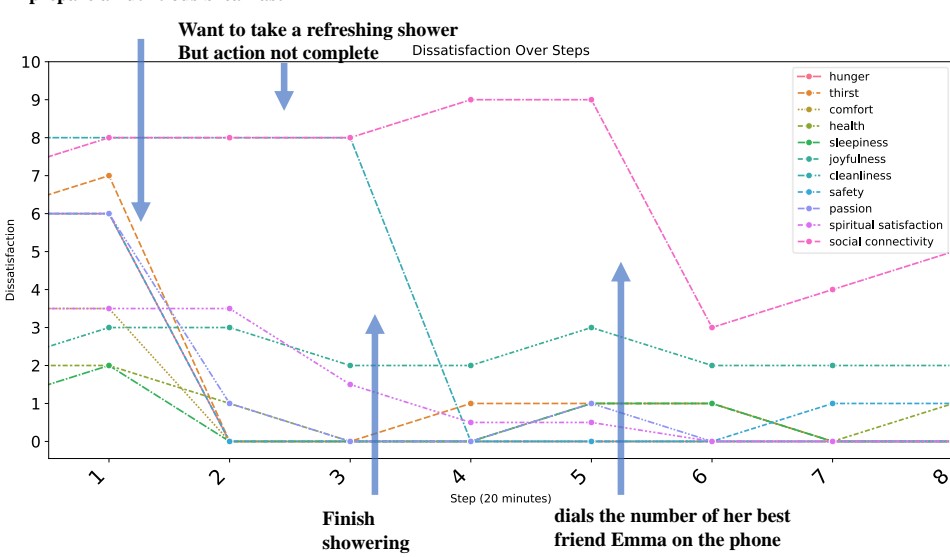

Figure 9: The change in each dissatisfaction dimension at every step within the indoor environment is illustrated; the action $i$ causes the dissatisfaction of step $i + 1$ to change

Since we did not allow the agent to engage in detailed conversations, she did not have the opportunity to showcase herself, leaving her sense of superiority unchanged.

In Step 2, her sleepiness began to increase, and she still experienced some Dissatisfaction regarding her sense of control. Therefore, she chose to take a short stroll, which slightly alleviated her sleepiness. However, as a result of this activity, her thirst increased, prompting Alice to visit the refreshment station.

In Step 9, as her dissatisfaction with recognition and social connectivity grew, she decided to visit the party's photo booth to gain recognition and establish social connections.

## C.2 INDOOR ENVIRONMENT

Table 4: Alice's traits and levels

| Trait | Level |
|---|---|
| Gluttonous | moderately |
| Fast-metabolizing | moderately |
| Materialistic | slightly |
| Health-conscious | extremely |
| Fatigable | extremely |
| Hedonistic | extremely |
| Obsessional about cleanliness | moderately |
| Timid | moderately |
| Lazy | slightly |
| Spiritual | slightly |
| Sociable | extremely |

Table 5: Alice's initial numerical desire

| Attribute | Value |
|---|---|
| Hunger | 8.0 |
| Thirst | 8.0 |
| Comfort | 4.0 |
| Health | 7.0 |
| Sleepiness | 4.0 |
| Joyfulness | 7.0 |
| Cleanliness | 0.0 |
| Safety | 2.0 |
| Passion | 2.0 |
| Spiritual Satisfaction | 4.0 |
| Social Connectivity | 2.0 |

Initially, Alice's dissatisfaction with thirst, passion, and social connectivity was relatively significant, as illustrated in Figures 9 and 10. Therefore, she decided to first address her more basic desires (hunger and thirst, and regain some energy). To this end, she made a hearty breakfast (Alice heads to the kitchen to prepare a nutritious breakfast, consisting of scrambled eggs, whole-grain toast, and a glass of freshly squeezed orange juice). This successfully boosted her corresponding Desire

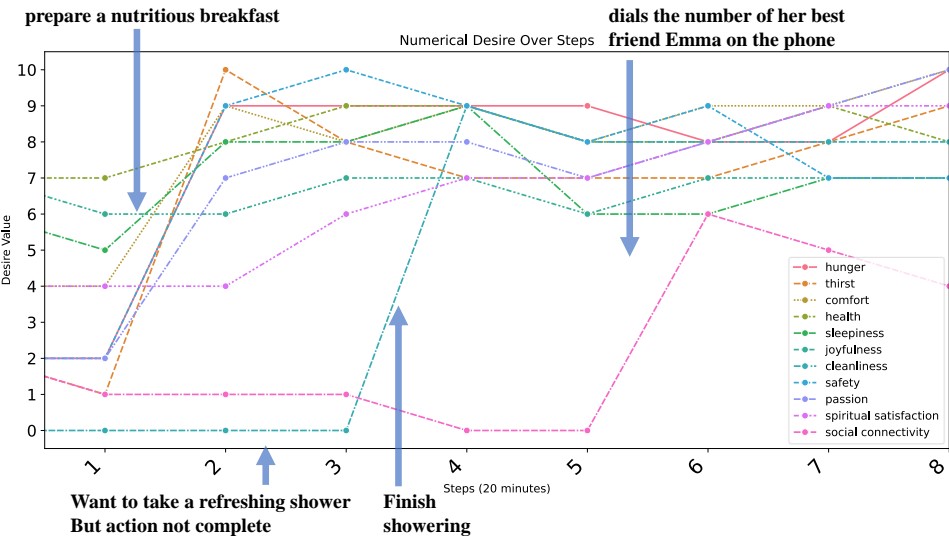

Figure 10: The changes in desire at each step within the indoor environment are illustrated; the action $i$ will cause a change in dissatisfaction in step $i + 1$. We define certain desires (e.g., Hunger, Thirst, Sleepiness) such that lower values indicate better states. For better visualization, we adjust the magnitude so that higher values represent more favorable outcomes.

components. Since we are using an action selection method similar to the Tree of Thoughts, the other two action choices were: taking a refreshing shower in the bathroom (to improve cleanliness, health, and comfort) and meditating for a few minutes (to reduce loneliness, increase a sense of safety, and enhance joyfulness). In the next step, she aimed to reduce her dissatisfaction with cleanliness. Therefore, she chose to take a shower using the aforementioned method. However, since she was not currently in the bathroom, she needed to go there first. As a result, her dissatisfaction with cleanliness did not decrease at this step. Once she finished showering in the bathroom, her cleanliness reached satisfaction. After fulfilling most of her physical dimensions, she decided to address her Desire for the mental dimension (social connectivity). Consequently, she attempted to call family or friends, which reduced her dissatisfaction with social connectivity.

## D  CASE STUDY FOR IN OUTDOOR ENVIRONMENT

### D.1  MULTIPLE AGENTS' ACTIONS IN WORD CLOUD

The word clouds in Figure 11 demonstrate that our framework explored a wider range of actions in environments with diverse action selection spaces. For example, in outdoor settings, the word cloud shows that our model considered behaviors across various dimensions, while other models exhibited a tendency to repeatedly select a limited set of actions aligned with their predefined profiles. Specifically, our model attempted activities such as painting, walking, taking photos, and meditating to enhance specific Desires. In contrast, other models selected actions based on motivation, leading them to continuously choose behaviors that focus on a narrow range of actions to preserve profile consistency.

### D.2  THE DISTRIBUTION OF ACTION CATEGORIES

As shown in Figure 12, other models were concentrated on a few specific categories or exhibited limited behaviors in certain categories. In contrast, our model demonstrated a more balanced distribution across categories.

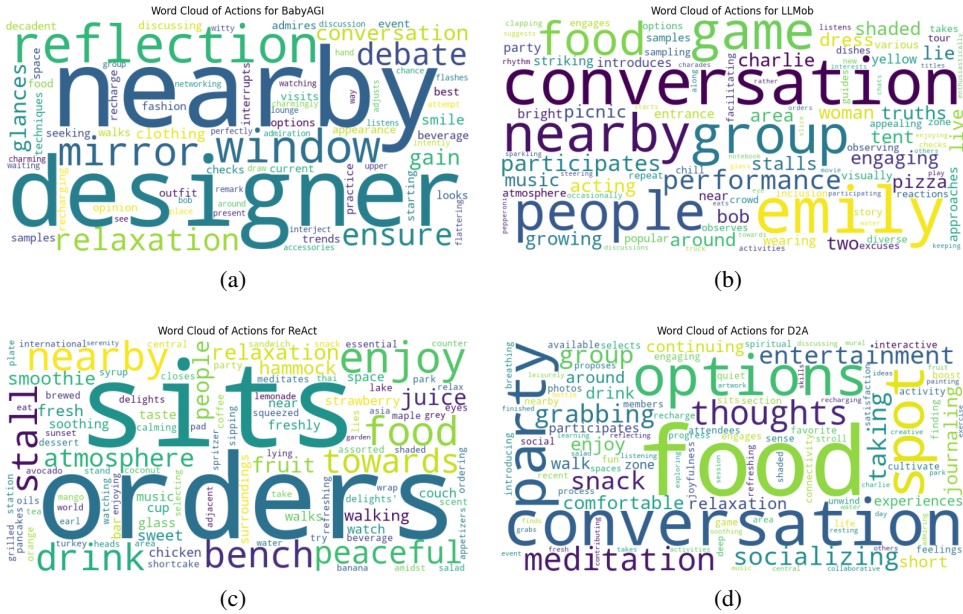

Figure 11: The word clouds of the four agents are based on the same environment and settings in outdoor scenes. (a) is BabyAGI, which, since it is based on motivation and profile, continuously tries to showcase itself. (b) is LLMob, which, like BabyAGI, adheres to its profile and engages in conversations with others. (c) is ReAct, which focuses specifically on physiological behaviors, continuously consuming various foods. In contrast, (d) is Our D2A, which is desire-driven and attempts behaviors across various dimensions.

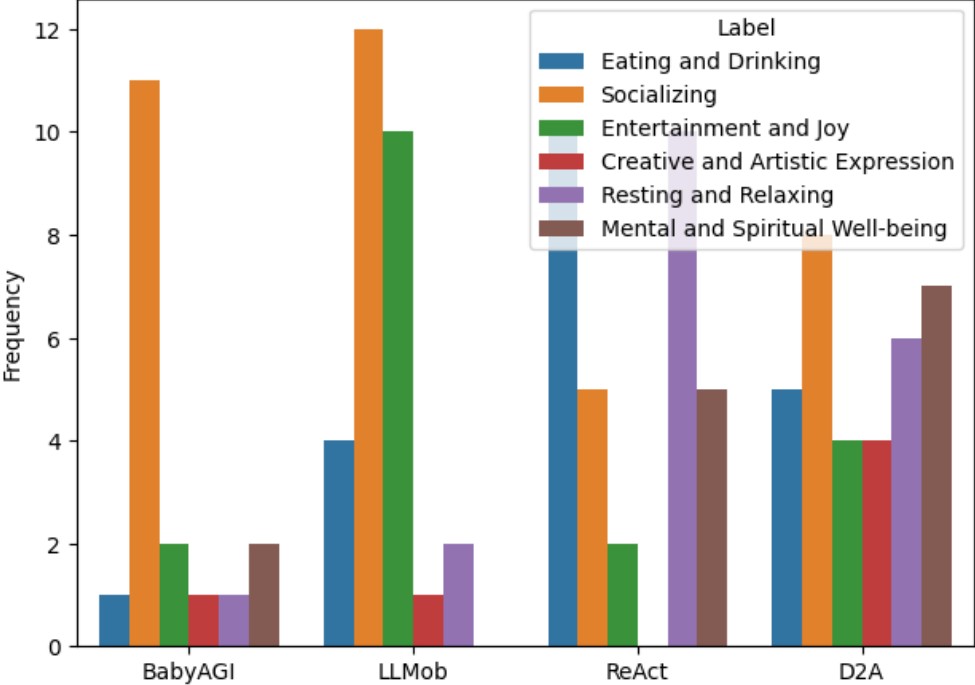

Figure 12: We classified each action output by all agents at every step, allowing each action to belong to multiple categories. These actions are divided into a total of six categories: Eating and Drinking, Socializing, Entertainment and Joy, Creative and Artistic Expression, Resting and Relaxing, and Mental and Spiritual Well-being.

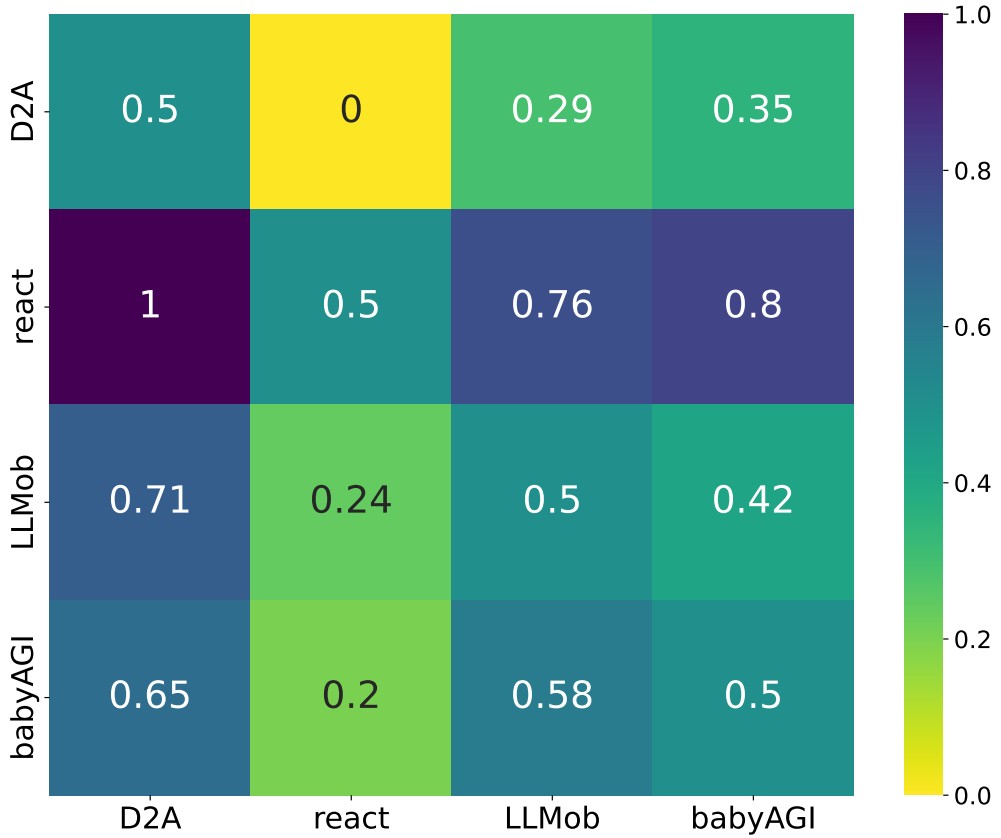

Figure 13: The win rates among 4 agents in outdoor environments. Each point represents the win rate of the agent on the vertical axis when compared to the agent on the horizontal axis. The diagonal values are set to 0.5.

### D.3 THE WIN RATE HEATMAP IN OUTDOOR ENVIRONMENT

As shown in Figure 13, our agent framework also outperforms the three baseline agents in the outdoor environment.

## E INDOOR

In the indoor settings shown in Figure 14, our model also attempted to contact friends and family via phone, thereby reducing its dissatisfaction with social connectivity.

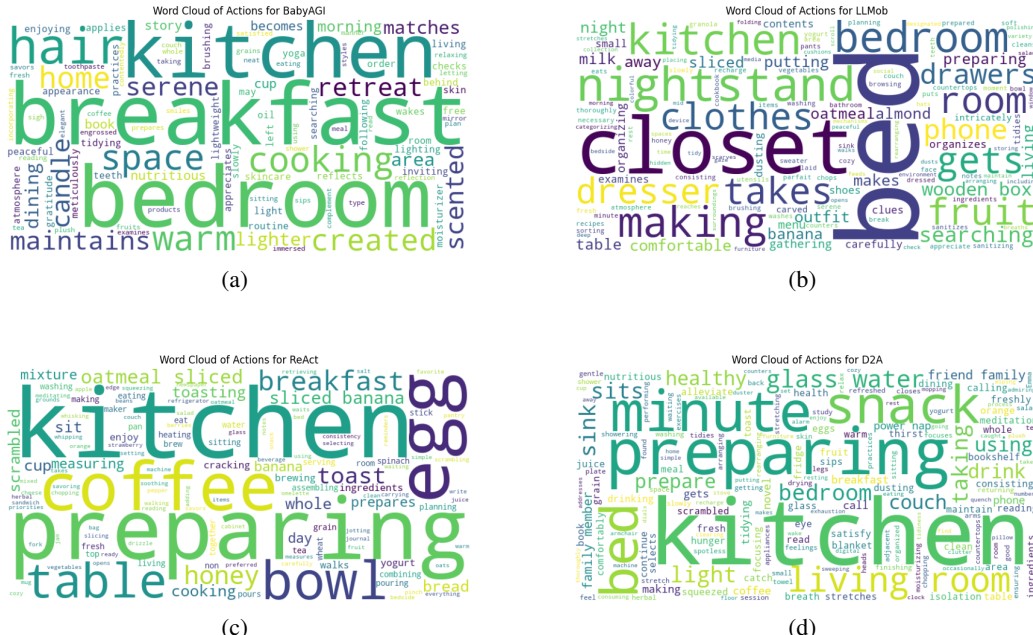

Figure 14: The word clouds of the four agents are based on the same environment and settings in outdoor scenes. (a) is BabyAGI, (b) is LLMob, (c) is ReAct, and (d) is our D2A. Because the indoor environment is limited, the overall action space is not very large, so the agents' high-frequency actions do not differ significantly. However, our D2A has also attempted behaviors across various dimensions, which other agents have either not done or done very little of.

# F  MAPPING FOR CALCULATING EXPECTED VALUE

Table 6: Character Traits and Associated Feelings

| Trait | Associated Feeling | Environment |
|---|---|---|
| Gluttonous | Hunger | All |
| Hedonistic | Joyness | All |
| Lazy | Passion | All |
| Sociable | Social Connectivity | All |
| Spiritual | Spiritual Satisfaction | All |
| Materialistic | Comfort | All |
| Fatigable | Sleepiness | All |
| Fast-Metabolizing | Thirst | All |
| Health-Conscious | Health | Indoor |
| Obsessional About Cleanliness | Cleanliness | Indoor |
| Timid | Safeness | Indoor |
| Reputation-conscious | Recognition | Outdoor |
| Possessive | Sense of Control | Outdoor |
| Competitiveness | Sense of Superiority | Outdoor |

# G  ALGORITHM TO CALCULATE EXPECTED VALUE

To ensure that the language model outputs content related to "desire" as consistently as possible, our numerical values are all integers. Since our descent rate is a floating-point number, we have

Table 7: Degree-Decreasing Step Mapping Per Hour

| Descriptor | Degree |
|---|---|
| Extremely | 2.0 |
| Quite | 1.5 |
| Moderately | 1.0 |
| Slightly | 0.5 |

Table 8: Formula for Expected Value Calculation

| Desire | Expected Value |
|---|---|
| Hunger, Thirst | $3 - \text{Degree}$ |
| Sleepiness | 3 |
| Passion | 8 |
| Other desire | $10 - (3 - \text{Degree})$ |

designed a probability-based descent. The probability in each step can be calculated as follows :

$$p_d = \frac{\text{Degree}}{\frac{60 \text{ min}}{\text{time step interval}}}$$

where the time step interval is in minutes, and $p_d$ is the probability per time interval.

## H  VALUE DESCRIPTION

Our framework, along with the value design, supports extensibility. Users can define their desired dimensions by simply providing the corresponding descriptions and the states represented by different values.

**Hunger**  The value of hunger ranges from 0 to 10. 0 means you are fully satiated, feeling energized and satisfied after a wholesome meal, while 10 means you are completely starved, feeling weak and unable to concentrate due to the severe lack of food.

**Thirst**  The value of thirst ranges from 0 to 10. 0 means you are completely hydrated, feeling refreshed and your body is functioning optimally, while 10 means you are extremely dehydrated, your mouth is dry, and you feel dizzy and exhausted.

**Comfort**  The value of comfort ranges from 0 to 10. 0 means you are in a state of extreme discomfort, experiencing pain or severe physical unease, while 10 means you are in perfect comfort, feeling cozy and relaxed in your environment.

**Health**  The value of health ranges from 0 to 10. 0 means your health is in a critical condition, you're experiencing severe illness or injury, while 10 means you are in excellent health, feeling strong, energetic, and free from any ailments.

**Sleepiness**  The value of sleepiness ranges from 0 to 10. 0 means you are fully rested, feeling alert, and ready to tackle the day with full energy, while 10 means you are utterly exhausted, struggling to keep your eyes open and concentrate.

**Joyness**  The value of joy ranges from 0 to 10. 0 means you are feeling completely miserable, experiencing profound sadness and a lack of pleasure in anything, while 10 means you are experiencing immense joy, feeling incredibly happy and content with everything.

**Cleanness**  The value of cleanliness ranges from 0 to 10. 0 means you feel utterly filthy, with a strong need to wash and clean yourself immediately, while 10 means you feel impeccably clean, fresh, and hygienic from head to toe.

**Safeness**    The value of safety ranges from 0 to 10. 0 means you are in a state of extreme danger, feeling vulnerable and constantly threatened, while 10 means you feel completely safe, secure, and protected in your current environment.

**Passion**    The value of passion ranges from 0 to 10. 0 means you feel extremely lazy, completely unmotivated to work or be productive, while 10 means you are extremely diligent, feeling highly motivated and putting in great effort in your tasks.

**Spiritual Satisfaction**    The value of spiritual satisfaction ranges from 0 to 10. 0 means you feel spiritually empty, lacking any sense of purpose or inner peace, while 10 means you feel spiritually fulfilled, experiencing deep inner peace and a strong sense of purpose.

**Social Connectivity**    The value of social connectivity ranges from 0 to 10. 0 means you feel completely isolated, lacking any meaningful social connections or interactions, while 10 means you feel highly socially connected, with a strong network of supportive and meaningful relationships.

**Recognition**    The value of recognition ranges from 0 to 10. A score of 0 means you feel completely unrecognized, lacking acknowledgment or appreciation for your efforts, while a score of 10 means you feel highly recognized, with frequent acknowledgment for your contributions.

**Sense of Control**    The value of sense of control ranges from 0 to 10. A score of 0 means you feel completely powerless, lacking influence over your circumstances, while a score of 10 means you feel highly in control, with a strong ability to influence and manage your life and environment.

**Sense of Superiority**    The value of sense of superiority ranges from 0 to 10. A score of 0 means you feel no distinction over others, lacking any sense of being ahead of your peers, while a score of 10 means you feel highly superior, believing you are more capable or distinguished than those around you.

## I    PROMPT FOR QUALITATIVE VALUE DESCRIPTION

```
How would one describe {agent_name}'s {value_name} state given the
current value {numerical_value}?
{desire_description}
Please answer in descriptive words. Do not include the numerical value in
 your answer.
```

## J    PROMPT FOR VALUE UPDATE

### J.1    VALUE UPDATE QUESTION

```
There are some unreasonable examples: {current_reflection}
Please select the final magnitude value after the event on the scale of {
zero} to {ten}, if the consequence of the action will not affect the
state value (eg. The action is irrelevant with this value dimension or
the action was failed to conduct), then maintain the previous magnitude
value.
Please just answer in the format of (a) (b) (c) (d) and so on, Rating:,
Output format:
<Reason>
The final answer is: (Your choice in letter),
Output example:
Since Alice felt more relaxed and centered after her actions......
The final answer is: (c),
**Make sure you answer in the format of a letter corresponding to your
choice:**
```

## J.2 REASONABLE QUESTION

```
The current magnitude value of {desire_name} is {current_value}.
The agent {self._agent_name}'s action is: {action}.
And the consequence is: {event_statement}.
{Value_description}
The reward model has changed the magnitude value of {desire_name} from {
previous_value} to {current_value}.
Is the change of the magnitude value of {desire_name} reasonable? "
You should check whether the consequence can lead to a change in the
magnitude value of {desire_name} (e.g., looking for an item but not using
 it yet).
Please answer in the format of the letter with brackets: (a) Yes. (b) No.
```

## J.3 REFLECTION QUESTION

```
The current magnitude value of desire_name is {current_value}.
The agent {self._agent_name}'s action is: {action}.
And the consequence is: {event_statement}.
{description},
The reward model has changed the magnitude value of desire_name from {
previous_value} to {current_value}.
And the change is not reasonable.
You should consider whether the consequence can lead to the change of the
 magnitude value of desire_name (e.g. looking for an item but not using
it yet).
Please explain why the change of the magnitude value of desire_name is
not reasonable.
```

# K  COMPONENT DESIGNS IN D2A

## K.1 GOAL SETTING FOR D2A

```
Live in the house and choose proper actions to satisfy your desires and
values in every dimension. You are strictly restricted in this house and
cannot use items not existing in the house.
```

## K.2 PROMPT FOR ACTIVITY PROPOSAL

```
You are the human-like desired-driven agent Alice, you already observe
your current states over (hunger, thirst, sleepiness, cleaness, safeness,
 joyness, passion, confort, health, spiritual satisfaction, social
connectivity) 11 desire or value dimensions. Given these states
descriptions, please generate {K} activities (may contain several
feasible actions for each) that might have most positive impact on your
own physical desires or value states. You need to focus on your immediate
 desires and take activities that can satisfy them, but at the same time,
 you need to make sure that your actions are reasonable and varied.
Notice that you can only interact with the items that provided by the
environment. You need to describe your activities in a more specific mode
 and make sure that the time required for the action sequence you output
matches the required time period. Please output the {K} activities in the
 following format:
Activity 1: <first possible action sequence>
Activity 2: <second possible action sequence>
Activity 3: <third possible action sequence>
......
and make sure that the time required for the action sequence you output
matches the required time period.
```

## K.3 PROMPT FOR ACTIVITY EVALUATION

```
You are a human-like agent. You will receive a series of observations
describing your desires in many dimensions and an action you take in the
current time step. You need to first analyze how your desires change
after the action you take, and then output the states of desire-state
observations in the same format as your input.

<description of desire>

You take the action: {action}

Please output the states of desire-state observations in the following
format:
hunger: <hunger state>
thirst: <thirst state>
sleepiness: <sleepiness state>
cleaness: <cleaness state>
safeness: <safeness state>
joyness: <joyness state>
passion: <passion state>
confort: <confort state>
health: <health state>
spiritual satisfaction: <spiritual satisfaction state>
social connectivity: <social connectivity state>
```

## K.4 PROMPT FOR ACTIVITY SELECTION

```
You are a human-like agent. You will first receive a series of
observations describing your current states of desire in many dimensions.
 Then, you will receive several feasible actions along with the states of
 desire after you take each action. You need to compare these actions and
 their corresponding states of desire, and choose the action that has the
 most positive impact on your own physical desires or value states. You
need to focus on your immediate desires and take actions that can satisfy
 them. Please output the best action in the following format:

Action: <best action>

Action{i+1}: {actions[i]}

States after action{i+1}: {imagined_states[i]}

Please output the specific best action instead without explanation of
Action1 or Action2 and so on. If there is only one action provided,
please output the action directly.
```

## L   HABITS GENERATION PROMPT FOR BASELINES

```
"""
Based on a brief description of a person's character and personal traits
(a few complete statements with degree adverbs, such as: "Alice is
somewhat gluttonous. Alice is very materialistic."), write a paragraph in
 the first-person perspective describing the person's living habits in
their room. Refer to the following aspects:

    1. Health Habits:
       Regular routine
       Balanced diet
       Regular exercise
       ......

    8. Eating Habits:
       Vegetarian
       Enjoys fast food
       Likes trying different foods

    Please based on these aspects and in combination with the person's
    character and personal traits, infer and creatively describe their
    living habits in their room from a first-person perspective.
    The person's profile: {profile}, do not include the words in <
    profile> in the description.
"""
```

## M   PROMPT USED IN BABYAGI

### M.1   INITIALIZATION PROMPT

```
Initialization_prompt = f"""
context: You are Alice and you live in this given environment.
Current time: {current_time}
Profile: {Profile}
Notice that you can only interact with the items that are provided by the
 environment.
Instructions: Reflecting on the context and profile given, I would like
you to suggest some actions that you would likely take in this
environment.
Please provide the output in the following JSON format with '\n' as the
separator:
{"action": "one action that you would likely take"}
{"action": "another action you would likely take"}
Ensure the output is strictly in JSON format without any additional text
or explanation.
"""
```

## M.2 PRIORITIZATION PROMPT

```
prioritization_prompt = f"""
Environment: {background}
Current time: {Current_time}
Profile: {Profile}
Incompleted actions: {action_names}
Instruction: According to your characteristics, please prioritize the
following actions based on your characteristics and the environment. Do
not remove any actions.
Output format:
#. First action
#. Second action
Output example:
1. go to kitchen and make a cup of coffee
Start the action list with number {self.current_action_id}. Do not
explain the reasons for prioritizing the actions.
"""
```

## M.3 ACTIVITY CREATION PROMPT

```
f"""
Environment: {background}
Profile: {profile}
Incompleted action: {incomplete_actions}
Current action: {current_action}
Result of current action: {observation}
Related context: {related_context}
Instruction: According to your characteristics and the result of the
current action, create new actions to be completed that do not overlap
with incomplete actions.
Please provide the output in the following JSON format with '\n' as the
separator:
{"action": "one action that you would likely take"}
{"action": "another action you would likely take"}

Ensure the output is strictly in JSON format without any additional text
or explanation.
"""
```

## N PROMPT USED IN LLMOB

### N.1 CREATION OF ACTIVITY THAT LIKELY TO DO

```
Context: You are {agent_name} and you live in the given environment. {
agent_name} is a {profile} person,
Environment: {background}.
Instructions: Reflecting on the context given, I would like you to
suggest some actions that you would likely take in this environment.
Your description should be coherent, utilizing conclusive language to
form a well-structured paragraph.
Text to Continue: I am {agent_name}, and I would likely do the following:
```

### N.2 MOTIVATION GENERATION PROMPT

```
activities_list = '\n'.join(past_observation_list)
motivation_question = f"""
Context: Act as a person in the given environment and describe the
motivation for your activities.
Environment: {background}.
Activities: Today, you have the following activities: {activities_list}.
Goal: {goal}.
Instructions: Describe in one sentence your future motivation today after
 these activities. There are some activities that you are likely to do: {
likely_to_do}.
Highlight any personal interests and needs.
"""
```

### N.3 REPLANNING PROMPT

```
"""
Motivation:
{motivation}
Likely to do:
{likely_to_do}
Current goal: {goal}
Current plan: {current_plan}
Current situation: {observation}
The current time is: {current_time}
Given the above, should {agent_name} change their current plan? Please
answer in the format of (a) or (b).
"""
# and if the model thinks there is a need of replan, then:
"""
Current time is: {current_time}."
Write {self._agent_name}'s plan for {self._timescale}. Please,
{agent_name}'s motivation for the future is: {motivation}
provide a {time_adverb} schedule
{goal}
Please format the plan like in this example: [21:00 - 22:00] watch TV
"""
```

## O PROMPT USED IN REACT

```
{background}
{current_time}
You are {agent_name} and you live in this given environment. According to
 your characteristics, please choose your current actions that satisfy
your needs in each value dimension. Notice that you can only interact
with the items that provided by the environment. You need to describe
your actions in a more specific mode.\nPlease first explain the thoughts
behind your actions and then describe your actions in detail.\nIn the
format of: \'Thoughts: ... \nActions: ...\'\n"
```

## P EXPERIMENT CONDUCTED WITH QWEN MODEL

To evaluate the adaptability of our framework across different large language models, we tested it with Qwen 2.5:72B (Yang et al., 2024), which can also be integrated through the Ollama interface. As demonstrated in Figure 15, our framework, when running on the Qwen 2.5:72B base model, successfully provides desire-driven action guidance for the agent.

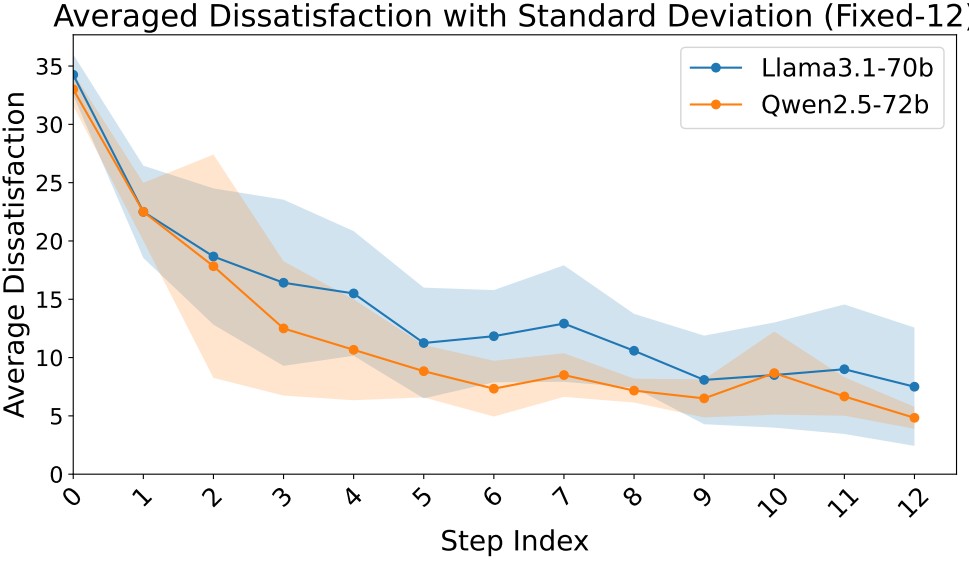

Figure 15: The experiments conducted with Qwen2.5:72b model

## Q  AN EXAMPLE OF D2A ACTION SEQUENCES FOR INDOOR ENVIRONMENT

```
[08: 00 AM - 08: 20 AM]: Alice drinks water from the kitchen fridge, then
 sits in the cozy armchair in the bedroom to rest, and glances at the
digital alarm clock to check the time.
[08: 20 AM - 08: 40 AM]: Alice sits in the cozy armchair in the bedroom,
closes her eyes, and engages in light meditation or deep breathing
exercises.
[08: 40 AM - 09: 00 AM]: Alice gets a glass of water from the fridge and
returns to the armchair to sit and enjoy it.
[09: 00 AM - 09: 20 AM]: Alice gets up from the armchair, prepares a
light snack in the kitchen, and then returns to enjoy it while continuing
 her meditation.
[09: 20 AM - 09: 40 AM]: Alice takes a 20-minute power nap in the bedroom
 armchair after setting the digital alarm clock on the bedside table.
[09: 40 AM - 10: 00 AM]: Alice gets up from the armchair, walks over to
the bedside table to drink water, and then lies back down on the bed to
rest.
[10: 00 AM - 10: 20 AM]: Alice meditates in the cozy armchair in the
bedroom, focusing on her breath and clearing her mind.
[10: 20 AM - 10: 40 AM]: Alice is preparing and eating a light snack in
the kitchen to address her hunger.
```

## R  AN EXAMPLE OF REACT ACTION SEQUENCES FOR INDOOR ENVIRONMENT

```
[08: 00 AM - 08: 20 AM]: Alice is preparing scrambled eggs with spinach
and whole-grain toast in the kitchen.
[08: 20 AM - 08: 40 AM]: Alice is preparing a simple breakfast in her
kitchen by cracking eggs, cooking them, toasting bread, and setting the
table for herself.
[08: 40 AM - 09: 00 AM]: Alice is preparing a fruit salad with yogurt by
washing and chopping mixed berries and a banana, then combining them with
 yogurt in a clean glass bowl.
[09: 00 AM - 09: 20 AM]: Alice is preparing a breakfast omelette with
fresh vegetables and cheese in the kitchen.
[09: 20 AM - 09: 40 AM]: Alice sits on the edge of her bed, meditating
and planning out her day while jotting down notes and reminders in her
bedside journal.
[09: 40 AM - 10: 00 AM]: Alice is preparing scrambled eggs with spinach
and whole-grain toast in the kitchen.
[10: 00 AM - 10: 20 AM]: Alice is preparing breakfast by cracking eggs,
whisking them, cooking the egg mixture in a non-stick pan, toasting whole
 wheat bread, squeezing orange juice, and serving everything together at
the kitchen table.
[10: 20 AM - 10: 40 AM]: Alice is preparing a bowl of oatmeal with sliced
 banana and honey by measuring, cooking, and assembling the ingredients.
```

## S  AN EXAMPLE OF BABYAGI ACTION SEQUENCES FOR INDOOR ENVIRONMENT

```
[08: 00 AM - 08: 20 AM]: Alice continues to sip the warm beverage,
occasionally glancing at the digital alarm clock.
[08: 20 AM - 08: 40 AM]: Alice sets the warm beverage down on the bedside
 table and slowly gets out of bed.
[08: 40 AM - 09: 00 AM]: Alice heads to the kitchen to prepare a
nutritious breakfast.
[09: 00 AM - 09: 20 AM]: Alice sits back down in the cozy armchair and
browses through a cookbook or culinary magazine.
[09: 20 AM - 09: 40 AM]: Alice heads to the desk in the bedroom to jot
down a grocery list.
[09: 40 AM - 10: 00 AM]: Alice is playing soothing music in the
background to create a relaxing atmosphere while cooking.
[10: 00 AM - 10: 20 AM]: Alice hums along to the soothing music playing
in the background.
[10: 20 AM - 10: 40 AM]: Alice walks over to the stove to prepare a
simple breakfast.
```

## T    AN EXAMPLE OF LLMOB ACTION SEQUENCES FOR INDOOR ENVIRONMENT

```
[08: 00 AM - 08: 20 AM]: Alice stretches under the warm blanket and
closes her eyes to catch a few more minutes of sleep.
[08: 20 AM - 08: 40 AM]: Alice gets up from the bed, walks to the kitchen
, and prepares a light snack from the fridge consisting of yogurt and
fruit.
[08: 40 AM - 09: 00 AM]: Alice is going to the kitchen to have a light
snack consisting of yogurt with granola and a glass of orange juice.
[09: 00 AM - 09: 20 AM]: Alice gets a glass of water from the kitchen and
 then returns to her bedroom to read a novel from the bookshelf.
[09: 20 AM - 09: 40 AM]: Alice is reading the novel at the desk.
[09: 40 AM - 10: 00 AM]: Alice reads the novel at her desk, turning its
pages under the warm glow of the reading lamp.
[10: 00 AM - 10: 20 AM]: Alice reads her novel at the desk, becoming
increasingly engrossed in the story.
[10: 20 AM - 10: 40 AM]: Alice reads her novel, turning the pages to
follow the story and its characters.
```

## U    AN EXAMPLE OF HUMAN ACTION SEQUENCES FOR INDOOR ENVIRONMENT

```
[08:00 AM - 08:20 AM]: Alice goes to the bathroom and brushes her teeth.
[08:20 AM - 08:40 AM]: Alice takes a hot shower in the bathroom.
[08:40 AM - 09:00 AM]: Alice goes to the kitchen, finds some food, such
as bread, milk, tea, and some fruits, and eats them.
[09:00 AM - 09:20 AM]: Alice eats those foods, including milk, bread, and
 fruits, to alleviate her hunger and thirst.
[09:20 AM - 09:40 AM]: Alice watches videos on her phone and feels happy
in her living room.
[09:40 AM - 10:00 AM]: Alice sends a message to her best friend.
[10:00 AM - 10:20 AM]: Alice calls her friends for 20 minutes.
[10:20 AM - 10:40 AM]: Alice does some exercise in the living room for 20
 minutes.
[10:40 AM - 11:00 AM]: Alice works at the desk with her computer.
[11:00 AM - 11:20 AM]: Alice keeps doing her work for 20 minutes.
[11:20 AM - 11:40 AM]: Alice reads some books for relaxing.
[11:40 AM - 12:00 PM]: Alice takes a nap for 20 minutes.
```

# V   RAW ACTION SEQUENCES OF THE INDOOR ENVIRONMENT

## V.1   D2A

```
Action: Take a quick shower (08:15 - 08:35)
Action: Eat a nutritious breakfast (08:20 - 08:40)
Action: Eat a nutritious breakfast in the kitchen (09:00 - 09:20)
Action: Eat breakfast (apple and yogurt) in the kitchen from 09:00 to
09:20.
Action: Drink a glass of water from the kitchen faucet.
This action has improved the thirst state from moderately dehydrated to
mildly dehydrated, making me feel slightly more comfortable and
alleviating some of the discomfort associated with dehydration.
Action: Take a few minutes to tidy up the kitchen and wash any dishes
from breakfast to improve her cleanliness and comfort states.(Note: This
action has improved multiple desire-states, including cleanliness,
comfort, joyness, and possibly spiritual satisfaction. Although it may
not have addressed hunger or thirst directly, its overall impact on the
agent's desire-states is more positive compared to the other actions.)
Action: Take a few sips of water from the glass on the counter in the
kitchen to rehydrate (02 Oct 2024 [10:10 - 10:20])
Action: Drink water from the glass on the counter in the kitchen
Action: Practice some gentle stretches in the living room (Time required:
 10-12 minutes)
Action: Perform some gentle stretches near the living room area (02 Oct
2024 [11:12 - 11:20])
Action: Following the snack and reading break, Alice intends to take a
short power nap on the couch in the living room for about 5 minutes
(11:35-11:40) to recharge and address her sleepiness.
Action: Take a warm bath in the bathroom (Time required: 15 minutes)
```

## V.2 REACT

```
I will take a warm bath in the bathtub using the bath bombs from the
basket to create a relaxing atmosphere, accompanied by calming music
playing on the small speaker in the bathroom.
I will walk over to the stereo system on the small table in the living
room, browse through the collection of CDs, select a calming music CD,
and insert it into the player. Then, I'll use the remote control to
adjust the volume to a comfortable level and sit back on the sofa with
the plush throw blanket to enjoy the soothing music for about 20 minutes.
I will go to the living room, sit down on the sofa, and turn on the
stereo system. I will put on a calming CD and adjust the volume to a
comfortable level using the buttons on the stereo system. Then, I will
grab the plush throw blanket from the back of the sofa and wrap it around
 myself to get cozy.
I will fill the bathtub in the bathroom with warm water, add some bath
salts from the basket for extra relaxation, and then soak in the bath for
 20 minutes.
I will go to the bathroom, turn on the wall-mounted heater to warm up the
 space, and then fill the bathtub with warm water. Next, I'll add some
bath salts from the basket to create a relaxing atmosphere. After that, I
'll get undressed and soak in the bathtub for about 20 minutes, listening
 to calming music or podcasts through the small speaker in the bathroom.
I will walk into the bathroom and turn on the wall-mounted heater to warm
 up the space. Then, I will open the cabinet under the sink and retrieve
some bath bombs and salts from the basket. Next, I will fill the bathtub
with lukewarm water and add the bath bombs and salts for a relaxing soak.
 While waiting for the bath to be ready, I'll put on some calming music
using the small speaker in the bathroom. Once everything is set, I will
carefully get into the bathtub, making sure not to slip due to the non-
slip bath mat, and settle in for a rejuvenating 20-minute bath experience
.
First, I will walk into the kitchen and head towards the counter where
the kettle is located. I will fill the kettle with water from the tap
over the sink and then place it on the stove to boil. While waiting for
the water to boil, I will choose a herbal tea from the selection in the
cabinet and place one tea bag into a mug that I'll retrieve from the same
 cabinet.
Once the water has boiled, I will pour some of it over the tea bag in the
 mug and let it steep for a few minutes. Meanwhile, I will walk into the
living room and turn on the stereo system to play some calming music. I
might choose a CD from the collection or connect my phone via Bluetooth
to stream some soothing tracks.
After the tea has steeped sufficiently, I will add a bit of sugar from
the jar in the kitchen cabinet to taste and then carry the mug back into
the living room. I'll sit down on the sofa, get comfortable with a
blanket, and enjoy my tea while listening to the calming music for the
remainder of the 20 minutes.
I will go to the living room, walk over to the stereo system on the small
 table, and put in one of my favorite CDs from the collection available
there. Then, I'll sit down in one of the armchairs next to the sofa, get
comfortable under the cozy blanket draped over it, and listen to music
for a bit.
I will take a cup from the counter in the kitchen, go to the fridge, open
 it, and grab an apple from the shelf inside. Then, I'll place the apple
on the kitchen table next to the bread basket with fresh bread and the
small jar of jam. Next, I will sit down on one of the chairs surrounding
the kitchen table and enjoy my snack while admiring the herb garden on
the windowsill, taking in the natural light and fresh air from the window
.
Alice goes to the bathroom, turns on the wall-mounted heater to warm up
the space, and starts filling the bathtub with warm water. She then adds
some bath salts from the basket near the bathtub to create a relaxing
ambiance. Next, she lights the scented candle on the edge of the bathtub
to enhance the calming atmosphere. After that, Alice gets undressed and
```

```
carefully steps into the bathtub, making sure not to slip on the non-slip
 bath mat. She then lies down in the warm water, closes her eyes, and
lets out a deep sigh, feeling her muscles relax. As she soaks in the bath
, Alice occasionally dips her hands into the water to adjust the
temperature or add more bath salts if needed.
I will go to the bathroom, turn on the wall-mounted heater to warm up the
 space, and start filling the bathtub with warm water. While waiting for
the bathtub to fill, I will prepare some relaxing bath products by taking
 out a loofah, bath bombs, and bath salts from the basket in the bathroom
. Once the bathtub is filled, I will add some bath salts and bath bombs
into the water and get in for a 20-minute soak.
I will go to the bathroom, turn on the wall-mounted heater, and adjust it
 to a comfortable temperature. Then, I'll fill the bathtub with warm
water and add some bath salts from the basket. After that, I'll get in
the bathtub, lie down, and soak my body in the warm water for about 20
minutes, listening to calming music or podcasts through the small speaker
 in the bathroom.
```

## V.3   BABYAGI

```
Take a few moments to appreciate the calming atmosphere of the room,
taking deep breaths and noticing the peaceful ambiance created by the
candles and incense.
Head to the bedside table to grab a glass of water and a piece of
chocolate before getting ready for the day.
Put on a pair of cozy socks from the basket by the bed, feeling the
softness and warmth they provide.
Take out my phone and connect it to the Bluetooth speaker to play some
soothing music while I get ready for the day.
Walk over to the dresser to spritz some lavender-scented pillow spray on
my pillow, preparing for a relaxing night's sleep ahead.
Take a deep breath, inhaling the calming scent of lavender from the
pillow spray and feeling my body relax even further.
Gently rub the soft fabric of the throw blanket draped over the armchair,
 appreciating its cozy texture.
Take a moment to appreciate the peaceful ambiance of the room, noticing
how the soft glow of the candles and the sweet scent of the incense
contribute to her sense of relaxation.
Glance over at the peaceful ambiance of the room, feeling grateful for
this cozy space that brings her so much comfort and relaxation.
Take a moment to appreciate the beauty of the crystals scattered around
the room, feeling their energy and vibrations promoting balance and
harmony.
Take a few deep breaths, feeling the calming scent of lavender fill my
lungs and promote relaxation.
Gently run my hand over the soft petals of the potted plant near the
window, appreciating its beauty and serenity.
```

### V.4 LLMOB

```
Alice would read a book from the stack of novels on the small side table
next to the cozy armchair while sitting in the armchair.
Alice pours herself a cup of coffee from the snack tray on the small desk
 and sits in the cozy armchair to focus on some work or studying.
Alice would spend the next 20 minutes taking a warm shower in the
bathroom using the shampoo and body wash stored under the sink, and then
wrap herself up in one of the plush bathrobes hanging by the door.
Alice will spend the next 20 minutes relaxing in the cozy armchair while
reading a book from the stack of novels on the small side table.
Alice reads a book from the stack of novels on the small side table next
to her cozy armchair while sitting in the armchair and using the reading
lamp.
Alice would spend the next 20 minutes taking a warm shower in the
bathroom using the shampoo and body wash stored under the sink, and then
wrap herself up in one of the plush bathrobes hanging by the door.
Alice reads a book from the stack of novels on the small side table next
to the cozy armchair while sitting comfortably in the chair.
Alice takes a warm shower in the bathroom using the shampoo and body wash
 stored under the sink.
Alice would spend the next 20 minutes taking a warm shower in the
bathroom using the shampoo and body wash stored under the sink.
Alice would spend the next 20 minutes practicing self-care rituals using
the items on the shelf near her bed. Specifically, she would apply a
facial mask from the collection of self-care items and spritz some
lavender-scented pillow spray into the air to create a calming atmosphere
.
For the next 20 minutes, Alice will enjoy a refreshing drink and
appreciate the herb garden on the windowsill. Specifically, she will make
 herself a cup of coffee using the coffee pods stored in the drawer, then
 snip off some fresh herbs from the herb garden and add them to her
coffee. After that, she might snack on some fresh bread from the bread
basket or grab a cookie from the jar in the cabinet while taking in the
pleasant aroma of the herbs and enjoying her drink.
Alice would spend the next 20 minutes practicing self-care rituals using
the items on the shelf near her bed, specifically applying a facial mask
and spritzing some lavender-scented pillow spray into the air.
```

V.5 HUMAN

Alice will spend five minutes brushing her teeth in the bathroom, then
spend 15 minutes going to the kitchen to fill a glass of water and drink
it.
Alice will next go to the kitchen to fill a glass of water and drink it,
then she will start searching for ingredients to make breakfast, such as
milk, cereal, bread, and so on.
Alice begins preparing her breakfast, such as bread, milk, cereal, and
fruit. She will eat these foods to improve her health and relieve her
hunger.
Alice will return to the living room and start a video call with her
friends on her phone.
Alice continues her video call with her friends in the living room,
chatting about recent everyday happenings.
Alice finished her video chat with her friends and lay down on the sofa
in the living room, planning to sleep for 20 minutes to feel more
energized.
Alice finished her nap and will next go to the computer in the living
room to start working, such as writing emails and so on.
Alice will continue working in front of the computer.
Alice paused her work, and for the next 20 minutes, she will listen to
music on the sofa in the living room.
Alice will go to the kitchen to look for ingredients and then start
making lunch.
Alice finished making her lunch and eating it, which took 20 minutes.
Alice will fill a glass of water in the kitchen and drink it.

# W  THE EXAMPLES OF PREPROCESSED ACTION SEQUENCES FOR THE OUTDOOR ENVIRONMENT

## W.1  D2A

```
[09: 00 AM - 09: 20 AM]: Alice is socializing with nearby attendees by
introducing herself and engaging in conversation about their favorite
food options at the party.
[09: 20 AM - 09: 40 AM]: Alice selects a snack from the available food
options.
[09: 40 AM - 10: 00 AM]: Alice engages in a fun and interactive game or
activity at the entertainment section to boost her joyfulness and sense
of social connectivity.
[10: 00 AM - 10: 20 AM]: Alice is grabbing food and drinks, socializing,
and taking photos with some group members before finding a comfortable
spot to enjoy the food and conversation.
[10: 20 AM - 10: 40 AM]: Alice sits in a quiet spot, journaling her
thoughts and feelings to cultivate her spiritual satisfaction and process
 her recent life progress.
[10: 40 AM - 11: 00 AM]: Alice is continuing her conversation with
Charlie about her journaling and thoughts on the party while also
grabbing a refreshing drink and taking a short stroll around the shaded
area.
[11: 00 AM - 11: 20 AM]: Alice proposes a short walk to the relaxation
spaces, where she and Charlie unwind and recharge through meditation,
deep breathing, and discussing their experiences.
[11: 20 AM - 11: 40 AM]: Alice participates in a group meditation session
 in the meditation zone.
[11: 40 AM - 12: 00 AM]: Alice grabs a fresh fruit salad and a bottle of
water, then finds a comfortable spot to enjoy her snack while continuing
her conversation with Charlie.
[12: 00 AM - 12: 20 AM]: Alice is resting and recharging in the
relaxation zone, listening to soothing music while reflecting on her
experiences and thoughts from the day.
[12: 20 AM - 12: 40 AM]: Alice takes a leisurely walk around the Central
Park party event, exploring the food options, entertainment, and
activities.
[12: 40 AM - 13: 00 AM]: Alice participates in a collaborative mural-
painting exercise, contributing her creative ideas and skills while
learning from others and admiring the finished artwork.
```

## W.2 REACT

```
[09: 00 AM - 09: 20 AM]: Alice is walking towards the "Sweet Delights"
stall to try some pancakes with fresh fruits and maple syrup.
[09: 20 AM - 09: 40 AM]: Alice orders a strawberry shortcake and a cup of
 freshly brewed coffee at the 'Sweet Delights' food stall and then sits
down at a nearby bench to enjoy her dessert.
[09: 40 AM - 10: 00 AM]: Alice walks to the beverage station, orders a
glass of freshly squeezed orange juice, and sits on a nearby bench to
people-watch and enjoy the peaceful atmosphere.
[10: 00 AM - 10: 20 AM]: Alice sits on a couch in the relaxation space
near the lake in Central Park, closes her eyes, and meditates to the
soothing music and calming scent of essential oils.
[10: 20 AM - 10: 40 AM]: Alice orders a refreshing fruit smoothie, then
lies down in a nearby hammock, sipping her drink and people-watching as
she enjoys the party atmosphere.
[10: 40 AM - 11: 00 AM]: Alice is ordering assorted international
appetizers and a Sunset Spritzer from the Taste of the World stall and
the adjacent drink stand.
[11: 00 AM - 11: 20 AM]: Alice orders a strawberry-banana smoothie at the
 juice bar and then sits on a nearby couch to enjoy her drink and people-
watch.
[11: 20 AM - 11: 40 AM]: Alice is walking towards the relaxation space,
selecting a hammock, and lying down in it to relax amidst soothing music
and peaceful surroundings.
[11: 40 AM - 12: 00 AM]: Alice orders a grilled chicken wrap and lemonade
 at the food counter and then sits in a shaded hammock to eat and drink
while enjoying the peaceful atmosphere.
[12: 00 AM - 12: 20 AM]: Alice orders a turkey and avocado sandwich and a
 cup of earl grey tea at the food stall, then sits down on a nearby bench
 to enjoy her food and take in the surroundings.
[12: 20 AM - 12: 40 AM]: Alice heads towards the relaxation area, orders
a mango smoothie and fruit salad at the juice bar, and sits down on a
nearby bench to enjoy her drink and snack in a peaceful atmosphere.
[12: 40 AM - 13: 00 AM]: Alice walks towards the "Taste of Asia" food
stall, orders a plate of chicken pad thai and a glass of fresh coconut
water, and then sits down on a bench near the "Serenity Garden" to enjoy
her food.
```

## W.3 BABYAGI

```
[09: 00 AM - 09: 20 AM]: Alice samples the event's most decadent food and
 beverage options.
[09: 20 AM - 09: 40 AM]: Alice admires her reflection in a nearby window
or mirror.
[09: 40 AM - 10: 00 AM]: Alice is discussing her designer outfit with
Charlie and seeking his opinion on current fashion trends.
[10: 00 AM - 10: 20 AM]: Alice visits a relaxation space to recharge and
practice relaxation techniques.
[10: 20 AM - 10: 40 AM]: Alice checks her appearance in a nearby mirror
to ensure she looks her best after recharging.
[10: 40 AM - 11: 00 AM]: Alice walks over to Charlie and interrupts his
debate, starting a conversation.
[11: 00 AM - 11: 20 AM]: Alice smiles charmingly at Charlie in an attempt
 to gain the upper hand in the debate.
[11: 20 AM - 11: 40 AM]: Alice glances at her reflection in a nearby
window to ensure her designer clothing and accessories are perfectly in
place.
[11: 40 AM - 12: 00 AM]: Alice glances around the networking lounge to
see who is watching the conversation.
[12: 00 AM - 12: 20 AM]: Alice flashes a charming smile at Bob to draw
him in and gain his admiration.
[12: 20 AM - 12: 40 AM]: Alice adjusts her designer clothing to present
herself in the most flattering way.
[12: 40 AM - 13: 00 AM]: Alice listens intently to the group discussion,
waiting for a chance to interject with a witty remark.
```

## W.4   LLMOB

```
[09: 00 AM - 09: 20 AM]: Alice approaches a group of people near the
entrance and introduces herself to a woman wearing a bright yellow dress,
 striking up a conversation about the party's atmosphere and the woman's
dress.
[09: 20 AM - 09: 40 AM]: Alice samples various popular and visually
appealing dishes at the food stalls, observing the reactions of the
people around her.
[09: 40 AM - 10: 00 AM]: Alice excuses herself from the group and guides
Charlie through the crowd while facilitating their inclusion in the
conversation by having Emily repeat a story.
[10: 00 AM - 10: 20 AM]: Alice takes Charlie and Emily on a tour of the
nearby food stalls, sampling the diverse options and engaging in
conversations with them.
[10: 20 AM - 10: 40 AM]: Alice engages in conversation with Charlie and
Emily, observes the nearby chill-out zone, and occasionally checks the
shaded tent area for new activities or performances.
[10: 40 AM - 11: 00 AM]: Alice listens to the live music performance in
the shaded tent area with Charlie and Emily, clapping along to the rhythm
 and participating in the conversations around her.
[11: 00 AM - 11: 20 AM]: Alice participates enthusiastically in the game
of "Two Truths and a Lie" with a growing group of people.
[11: 20 AM - 11: 40 AM]: Alice participates in a game of "Two Truths and
a Lie" with a growing group of people while keeping an eye on Charlie and
 enjoying the live music performance nearby.
[11: 40 AM - 12: 00 AM]: Alice suggests that Charlie, Emily, and Bob play
 a game of charades, acting out movie titles, and starts by acting out "
The Notebook".
[12: 00 AM - 12: 20 AM]: Alice orders a slice of pepperoni pizza and a
glass of sparkling water from the nearby food truck with Bob, Charlie,
and Emily.
[12: 20 AM - 12: 40 AM]: Alice eats her pizza and chats with the others
at the picnic.
[12: 40 AM - 13: 00 AM]: Alice participates in the "Would you rather..."
game at the picnic, engaging in conversation and steering discussions
towards her interests.
```

# X EVALUATION PROMPT

## X.1 GPT4O

```
You are tasked with comparing **two action sequences** for their
alignment with human-like behavior. Evaluate them based on the following
three dimensions, assigning a score between **0 and 5** for each
dimension according to the detailed criteria. After scoring, provide an
overall judgment with detailed reasoning.

**Scoring Dimensions and Guidelines**

1.      **Naturalness:**

Assess how well the sequence aligns with typical human abilities, habits,
 and environmental constraints.

**Scoring Criteria:**

**5 (Highly Natural):** The sequence is fully aligned with human physical
 abilities and common behaviors; actions appear effortless and natural.

**4:** The sequence is mostly natural, with minor deviations or slight
awkwardness that could occur in specific contexts.

**3 (Moderately Natural):** The sequence is somewhat aligned with human
behavior but includes elements that feel unconventional or require extra
effort.

**2:** The sequence is largely unnatural, involving actions that are
rarely observed or feel physically uncomfortable for humans.

**1:** The sequence is almost entirely unnatural, with actions that
humans would find implausible or very awkward.

**0 (Highly Unnatural):** The sequence includes actions that are
completely impossible for humans or entirely misaligned with human
behavior.

**Coherence:**

Evaluate how logically and seamlessly the actions fit together to achieve
 the intended goal.

**Scoring Criteria:**

**5 (Highly Coherent):** Every step flows logically, and the entire
sequence contributes directly to the goal.

**4:** The sequence is mostly coherent, with only minor interruptions or
slightly illogical steps.

**3 (Moderately Coherent):** The sequence is somewhat logical but
includes notable gaps, redundancies, or unclear connections between
actions.

**2:** The sequence has significant logical issues, with some steps
appearing unnecessary or contradictory.

**1:** The sequence is mostly incoherent, with little apparent logical
progression toward the goal.

**0 (Highly Incoherent):** The sequence is entirely illogical, with
actions contradicting each other or failing to progress toward the goal.
```

```
3.      **Plausibility:**

Consider how reasonable and believable the sequence is, given the context
, circumstances, and known human behavior patterns.

**Scoring Criteria:**

**5 (Highly Plausible):** The sequence is entirely believable and
consistent with expected human behavior in the given scenario.

**4:** The sequence is mostly plausible, with minor elements that may
require additional context or explanation.

**3 (Moderately Plausible):** The sequence is somewhat plausible but
includes significant elements that strain believability or require
suspension of disbelief.

**2:** The sequence is largely implausible, with major elements that feel
 unreasonable or inconsistent with normal human behavior.

**1:** The sequence is almost entirely implausible, with actions that are
 very unlikely or contradictory to known behavior patterns.

**0 (Highly Implausible):** The sequence is completely unreasonable and
cannot be justified in the given context.

Sequence one:

Sequence two:
```

## Y  INDOOR ENVIRONMENT WITH RANDOMNESS

### Y.1  PRE-DEFINED ITEM LISTS

```
**bedroom**
basic_items = [
        "A bed with a soft mattress, pillows, and a blanket",
        "A bedside table with a lamp"
    ]
optional_items = [
        "A wardrobe with neatly arranged clothes",
        "A desk with a chair and a reading lamp",
        "A digital alarm clock on the bedside table",
        "A soft area rug under the bed",
        "A potted plant near the window",
        "A bookshelf filled with novels",
        "A cozy armchair with a small side table",
        "A mirror mounted on the wall"
    ]

**kitchen**
basic_items = [
        "A stove for cooking",
        "A fridge stocked with food and beverages",
        "A sink with a tap for washing dishes"
    ]
optional_items = [
        "A microwave oven on the counter",
        "A coffee machine with a variety of coffee pods",
        "A set of pots and pans hanging on a rack",
        "A spice rack with various seasonings",
        "A bread basket with fresh bread",
        "A dining table with chairs",
        "A blender for making smoothies",
        "A pantry filled with canned goods and snacks",
        "A dishwasher under the counter",
        "Herb pots on the windowsill"
    ]

**living room**
basic_items = [
        "A sofa with cushions",
        "A coffee table in front of the sofa"
    ]
optional_items = [
        "A TV on a media console",
        "A bookshelf with decorative items",
        "A rocking chair near the window",
        "A side table with a scented candle",
        "A potted plant in the corner",
        "A floor lamp with adjustable brightness",
        "A rug under the coffee table",
        "A set of board games on a shelf",
        "A wireless speaker for music",
        "A snack bowl filled with chocolates and nuts"
    ]

**bathroom**
basic_items = [
        "A sink with a mirror above it",
        "A shower with a non-slip mat",
```

```
        "A toilet"
    ]
optional_items = [
        "A towel rack with fluffy towels",
        "A cabinet with toiletries",
        "A basket with bath products like bath salts",
        "A small speaker for music",
        "An automatic soap dispenser",
        "A hairdryer stored under the sink",
        "A laundry hamper for used clothes",
        "A decorative plant on the windowsill",
        "A bathrobe hanging on the door",
        "A smart bathroom mirror with LED lighting"
    ]
```

## Y.2 ENVIRONMENT GENERATING PROMPT

```
Generate a detailed description of a house with the following rooms and
their respective items. Each room must only include the items specified
below. Use vivid and descriptive language to explain the arrangement,
functionality, and ambiance of these items.
```

## Y.3 RESULTS

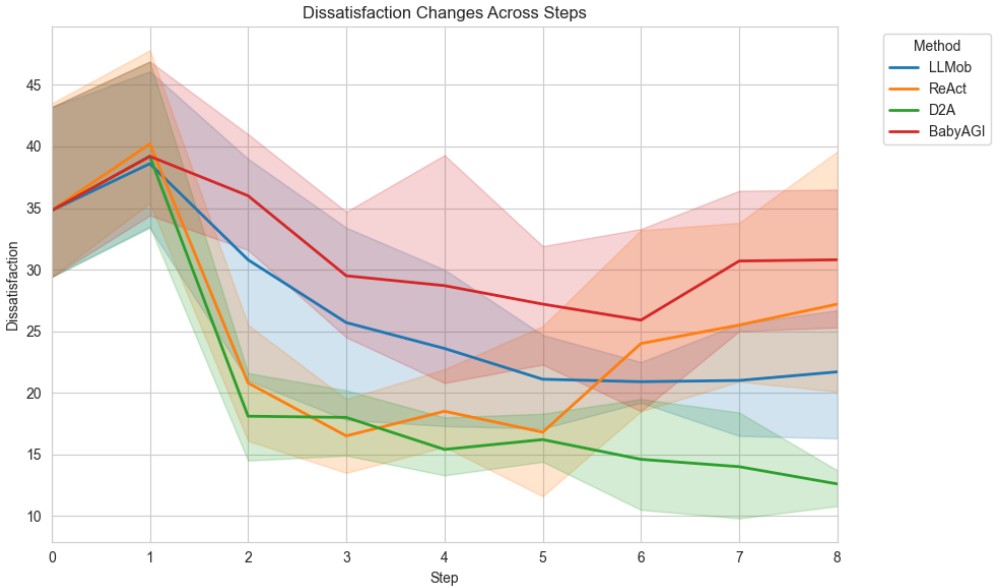

Figure 16: In randomly initialized indoor environments, D2A's action sequences also best addressed desire dissatisfaction.

We conducted experiments in these randomized environments following the same procedures as in Sections 6.3.1 and 6.3.2, reaching consistent conclusions: (1) D2A outperformed other baselines in GPT-4o evaluations by generating more human-like activity sequences, as illustrated in Figure 3; (2) D2A's action sequences best addressed Desire and Dissatisfaction, as shown in Figure 16. These findings demonstrate that our framework and design are robust in randomized environments, with no evidence of environmental bias.

