# OpenReview forum: "Simulating Human-like Daily Activities with Desire-driven Autonomy"
_ICLR.cc/2025/Conference — ICLR 2025 Poster_

### Official Review · Reviewer_dH5e · 2024-10-28

**Soundness:** 3
**Presentation:** 3
**Contribution:** 3
**Rating:** 6
**Confidence:** 3

**Summary:**

The paper introduces the Desire-driven Autonomous Agent (D2A), a framework for simulating human-like daily activities based on intrinsic motivations rather than explicit tasks or external rewards. Inspired by Maslow's Theory of Needs, D2A prioritizes actions that fulfill a hierarchy of desires (e.g., physiological, social, self-actualization), allowing it to autonomously select actions that align with its motivational framework. This desire-driven approach contrasts with traditional task-oriented agents by focusing on fulfilling internal motivations, which provides the agent with the capacity for more adaptable and human-like behavior.

**Strengths:**

**1. Proactive Action Based on Intrinsic Motivation**: The proposed D2A framework demonstrates an ability to proactively initiate actions driven by intrinsic motivations. Through the integration of a value system and a desire-driven planner, the framework establishes a dynamic interaction between desires and actions, wherein lowered desire values trigger corresponding activities to restore balance. This mechanism, though relatively simple, allows the agent to engage autonomously in daily activities in a manner that mirrors proactive human behavior, setting it apart from purely reactive or task-driven models.

**2. Human-Inspired Intrinsic Motivations Across Life Dimensions**: Unlike recent approaches that focus on intrinsic motivations for exploration or collaboration, the D2A framework offers a multi-dimensional model inspired by human needs. By integrating eleven desire dimensions (e.g., physiological, social, and self-fulfillment needs), D2A provides a broader, more human-like motivational structure. This approach goes beyond typical reward-driven or exploratory motivations by simulating daily life activities that dynamically balance internal desires, reflecting human motivation patterns more authentically. This novelty enhances the agent's potential for replicating realistic, human-inspired behaviors within single-agent environments.

**Weaknesses:**

**1. Inconsistent Application of Maslow’s Theory**: While D2A is presented as being inspired by Maslow’s Theory of Needs, it does not implement the theory’s hierarchical structure. According to Maslow, higher-level desires are pursued only once lower-level needs are met, but D2A treats each desire independently, allowing the agent to pursue higher-level needs without satisfying foundational ones. This weakens the theoretical foundation and could make the agent's behavior feel less authentically human-like.

**2. Overly Simplistic Desire-Action Dynamics**: Despite its innovative multi-dimensional motivational structure, the D2A framework uses a straightforward linear deduction of desire values to simulate the fluctuation of needs. This approach falls short of capturing the organic variation in human desires, which often intensify or wane in response to context, time, or recent actions. By exploring more complex dynamics—such as non-linear decay, situational adjustments, or time-of-day cycles—the framework could better reflect realistic interactions between desires and actions. The current simplicity reduces the authenticity of the agent's behavior, limiting the depth of its desire-driven model.

**3. Narrow Focus on Physiological Desires in Single-Agent Setting**: The experimental results in the paper primarily emphasize physiological desires, with minimal exploration of higher-level motivations such as social connectivity or self-actualization. Additionally, the study tests only one agent, limiting insights into potential interactions or complex social behaviors that might arise in multi-agent settings. These restrictions make the results narrowly focused and reduce the paper's ability to demonstrate the full range of behaviors that D2A could potentially simulate, ultimately constraining the framework’s demonstrated impact.

**4. Opaque GPT-4o Evaluation Methodology**: The evaluation of human-likeness using GPT-4o lacks transparency. Key details—such as the prompts used, scoring consistency, and validation of the assessment criteria—are not provided, making it difficult to gauge the robustness of the results. This lack of methodological clarity limits confidence in whether the evaluation effectively captures nuanced human-like behavior or merely reflects surface-level patterns.

**5. Limited Qualitative Evaluation of Generated Action Sequences**: The paper lacks an in-depth qualitative analysis of the action sequences generated by D2A, making it difficult to assess the framework’s true effectiveness. The brief, simplistic sequence shown in Appendix O does not illustrate the model’s potential for creating complex, dynamic routines, leaving the impact of D2A’s design unclear. More detailed, varied sample sequences, along with thoughtful discussion, would provide stronger evidence of D2A's capabilities in simulating realistic human-like behaviors.

**Questions:**

Please answer the issues mentioned in the weaknesses.

---

> ### Author Response · Authors · 2024-11-25
> **Response to Reviewer dH5e (1)**
>
> Thank you for your detailed and thoughtful review of our work. We appreciate your recognition of the novelty of the D2A framework, particularly its proactive action mechanism based on intrinsic motivations and the human-inspired multi-dimensional desire model. Your comments highlight several areas for improvement, and we have carefully addressed each point below.
>
> >**W1**: Inconsistent application of Maslow’s Theory.
>
> **R1**: We appreciate your thoughtful comments on the theoretical alignment and desire-action dynamics in our framework. Below, we address these concerns in detail:
>
> - Our framework design is inspired by the **theory of needs**, where the construction of value dimensions is not strictly bound by a predefined hierarchy such as Maslow’s. While we referenced Maslow’s Theory of Needs as inspiration, we believe there is **no universal standard for structuring these dimensions**. To demonstrate the flexibility and scalability of our framework, we introduced a new outdoor social scenario in our expanded experiments, where we replaced some of the original desire dimensions with alternative ones tailored to the new setting. The consistent effectiveness of the framework in guiding action generation across these different scenarios underscores its adaptability and robustness.
> - We acknowledge that certain simplifications were made in our current implementation. However, the **strong performance of our simplified model underscores the validity and importance of our paradigm**. It demonstrates that even with these simplifications, the intrinsic motivation framework effectively guides agent behaviors, showcasing the robustness of our approach. Exploring more complex mental hierarchies will be a key focus of our future work. We are confident that incorporating more sophisticated intrinsic motivational architectures, such as hierarchical relationships and non-linear interactions, will not only improve performance but also enhance the realism of agent behaviors.
>
> We improved our paper writing in the revision to avoid such misunderstanding of our motivations.
>
> >**W2**: Overly simplistic desire-action dynamics.
>
> **R2**: We appreciate your observation regarding the simplicity of the desire-action dynamics in our framework. Similar to our response to the previous point, our current implementation focuses on a simplified dynamic model that prioritizes computational efficiency. Despite its simplicity, the model has demonstrated strong performance in driving diverse and coherent activities, indicating the effectiveness of even a straightforward approach in guiding agent behavior.
>
> At the same time, we fully recognize the potential for enhancing the realism and depth of the framework by incorporating more complex dynamics. Future work will explore advanced modeling techniques, such as non-linear decay, situational adjustments, and time-of-day cycles. We are confident that these enhancements will further improve performance and allow the framework to better capture the nuanced variations in human desires, resulting in even more authentic and human-like behaviors.

---

> > ### Comment · Reviewer_dH5e · 2024-11-25
> >
> > I concur that the authors have provided appropriate responses to my comments. Regarding W1, the authors have revised the main text to clarify that their approach does not strictly adhere to Maslow's Theory, which makes it understandable that the hierarchical nature of needs in the theory is not directly considered. This explanation addresses my concern effectively.
> >
> > For W2, I find it reasonable at this stage to acknowledge the potential for extending the approach with more complex dynamics while leaving such enhancements for future work. I appreciate that, despite its simplicity, the current model offers valuable insights into the effectiveness of the D2A framework.

---

> ### Author Response · Authors · 2024-11-25
> **Response to Reviewer dH5e (2)**
>
> >**W3**: Narrow focus on physiological desires in a single-agent setting.
>
> **R3**: To address this concern, we have expanded our experiments to include scenarios that explore higher-level motivations and multi-agent interactions, showcasing the versatility and broader applicability of our framework.
>
> To better evaluate higher-level motivations, we introduced an **outdoor park party scenario** (see Appendix `A.2`), where agents interact in a rich, multi-agent social environment. In this scenario, we updated some of the desire dimensions to focus on higher-level motivations. The updated desire dimensions include **hunger, joyfulness, passion, social connectivity, health, spiritual satisfaction, comfort, cleanliness, sleepiness, safety, thirst, recognition, sense of control,** and **sense of superiority**. These refined dimensions allowed us to evaluate the agent’s ability to generate activities that align with complex and socially rich motivations, demonstrating the flexibility of our framework in adapting to different settings and needs.
>
> Additionally, to address the single-agent limitation, the outdoor park party scenario features **multi-agent interactions** with NPCs (see Appendix `A.3`), enabling the agent to engage in conversations and collaborative activities. This multi-agent environment provided valuable insights into the framework’s ability to simulate social behaviors and interactions, further broadening the range of human-like activities D2A can generate. Due to time constraints, we were unable to generate a large amount of experimental data in the new environment for a reliable comparison of human likeness. However, we conducted a statistical analysis on some of the action sequences obtained from the indoor and newly added outdoor scenarios. We first visualized the activity sequences obtained from the indoor and outdoor multi-agent scenarios using point cloud plots. Next, we defined several statistical labels to categorize and count the actions generated by each agent in the outdoor scenario. The results of these two statistical experiments demonstrate that our D2A can produce **diverse and balanced** actions in both indoor and outdoor scenarios. (More details in Appendix`D`.) The **consistent statistical conclusions** obtained here, along with **the reasonable activity sequences in the multi-agent environment**, demonstrate the scalability of our approach and its potential for simulating diverse, dynamic scenarios beyond single-agent settings.
>
> >**W4**: Opaque GPT-4o evaluation methodology.
>
> **R4**: It was an oversight not to include the prompt used for GPT-4o evaluations in the initial version of the paper; it has now been added to appendix `Y`. Specifically, GPT-4o evaluates each sequence by assigning scores (out of five) for the three dimensions of naturalness, coherence, and plausibility, and then aggregates these scores to determine which sequence appears more human-like.
>
> To validate the evaluation of GPT-4o, we randomly selected 19 pairs of activity sequences for pairwise comparison (16 pairs from the indoor scenarios and 3 pairs from the newly added outdoor scenarios) and created three questionnaires, each assigned to 8 human annotators. The human annotators were asked to judge which sequence in each pair appeared more human-like. Based on the level of agreement among the annotators, we categorized the 19 samples into three groups:
>
> - **High consensus**: Over 75% agreement on one side—indicating strong human consensus.
> - **Moderate preference**: Agreement between 50.1%–74.9%—indicating moderate human preference.
> - **Difficult to distinguish**: Exactly 50% agreement—indicating humans found it difficult to decide between the two sequences.
>
> We also provided the same pairs to GPT-4o, using our evaluation prompt and both sequences as context, asking it to determine which sequence appeared more human-like. The consistency rate between human evaluations and GPT-4o’s assessments is summarized in the table below:
>
> | **Human Annotator Agreement**       | **Proportion** | **GPT-4o Consistency Rate (%)** |
> |-------------------------------------|----------------|----------------------------------|
> | High consensus ($>75\%$ agreement) | 11/19          | 100                              |
> | Moderate preference ($50.1\%$–$74.9\%$) | 5/19           | 80.0                             |
> | Difficult to distinguish ($50\%$ agreement) | 3/19           | 66.7                             |
>
> Note that for the three pairs in the table where humans found it difficult to distinguish, GPT-4o also outputs a `draw` result for two of them.
> This verification demonstrates the robustness of our evaluation framework and **confirms that GPT-4o’s assessments are reliable and align closely with human judgments.**

---

> > ### Comment · Reviewer_dH5e · 2024-11-25
> >
> > Regarding W3 and W4, the authors have adequately addressed my concerns by providing additional experiments and detailed explanations. I appreciate the effort put into these revisions, which significantly strengthen the paper.

---

> ### Author Response · Authors · 2024-11-25
> **Response to Reviewer dH5e (3)**
>
> >**W5**: Limited qualitative evaluation of generated action sequences.
>
> To address the fifth issue, we conducted some statistical analysis to evaluate the **diversity and balance** of actions generated by different methods. Specifically, we first visualized the activity sequences obtained from the indoor and outdoor multi-agent scenarios using point cloud plots. Next, we defined several statistical labels to categorize and count the actions generated by each agent in the outdoor scenario.
>
> The word cloud figures (see Appendix `D.1`) emphasize that **our framework is capable of exploring a broader range of actions** in environments with **diverse action selection spaces**. For example, in outdoor settings, the word cloud reveals that our model considers behaviors spanning various dimensions, whereas other models tend to focus on a narrower set of actions within these dimensions. Specifically, our model engages in activities such as painting, walking, taking photos, and meditating to fulfill specific desires. In contrast, other models rely heavily on motivation-driven action selection, often repeatedly choosing behaviors that strictly align with their profiles, thereby limiting the diversity of their actions.
>
> The label-counting results showed that D2A-generated actions were **more evenly distributed across categories**, whereas the baselines displayed more extreme distributions—some with a heavy concentration in certain categories and others completely lacking actions in specific categories. This analysis highlights that our method can generate more diverse and balanced action sequences, making it comparatively more reasonable and adaptable than methods with extreme or unbalanced distributions.
>
> Additionally, we have included **a case study in `Appendix C` that analyzes how desires guide the generation of actions**, providing a detailed explanation of the underlying decision-making process. Meanwhile, `Appendix W and X` presents more examples of action sequences generated by D2A across various scenarios, offering a more intuitive understanding of the results and showcasing the diversity and quality of the generated activities.

---

> > ### Comment · Reviewer_dH5e · 2024-11-25
> >
> > Regarding W5:
> > Although the experiments remain limited in both quantity and diversity, the additional experiments and analysis provide evidence of the proposed model's effectiveness. However, to comprehensively evaluate the realism of virtual human models' activities, it is essential to establish a robust benchmark and well-defined metrics. These steps will be critical for assessing and comparing the performance of such models in future studies.

---

> ### Comment · Reviewer_dH5e · 2024-11-25
>
> The authors have addressed my concerns appropriately, and I have updated my score accordingly. I appreciate the authors' efforts in revising the manuscript and responding to the feedback thoroughly.

---

> > ### Author Response · Authors · 2024-11-25
> > **Thanks for your postive feedbacks**
> >
> > We would like to extend our appreciation for your positive feedback and score update.
> >
> > We are pleased to hear that our responses to your initial reviews, particularly regarding W1, W2, W3, and W4, have been found satisfactory. Regarding W5, we agree that it is important to establish robust benchmarks and well-defined metrics for the comprehensive evaluation of virtual human models. Your guidance on this matter is crucial for the future direction of our research.
> >
> > Once again, thank you for your time and your thoughtful review. Your kindness and insights will encourage us to continuously contribute to the field.

---

### Official Review · Reviewer_UXrD · 2024-10-30

**Soundness:** 2
**Presentation:** 4
**Contribution:** 2
**Rating:** 6
**Confidence:** 2

**Summary:**

The paper introduces a Desire-driven Autonomy framework for LLM-based agents to simulate human-like daily activities. The framework is inspired by Maslow's theory of needs and includes 11 dimensions of human-like desires. The Desire-driven Autonomous Agent (D2A) operates based on intrinsic motivations, autonomously proposing and selecting tasks that fulfill its motivational framework. The authors developed a flexible text-based activity simulator using Concordia components, supporting various agent types and textual environments for reliable interaction and evaluation. They conducted simulations in a detailed textual home environment with a Game Master providing relevant observations. The experiments demonstrated that D2A generates appropriate activities effectively and efficiently, achieving described desires. A comparative analysis with three baseline approaches (ReAct, BabyAGI, and LLMob) showed that D2A generates more natural, coherent, and plausible daily activities.

**Strengths:**

- The proposed framework allows agents to operate based on intrinsic motivations, which is a significant departure from existing task-oriented AI agents that rely on explicit instructions or external rewards.
- The paper is well written and has nice figures.
- The authors conducted a comprehensive comparative analysis with three baseline approaches (ReAct, BabyAGI, and LLMob) to evaluate the effectiveness of their framework. The development of a flexible text-based activity simulator using Concordia components is another strength of the paper.

**Weaknesses:**

I think there are some weaknesses in this paper:
- Limited Technological Innovation. The paper primarily focuses on the conceptual framework and theoretical underpinnings of the desire-driven autonomy approach. While the idea of using intrinsic motivations inspired by Maslow's theory of needs is innovative, the technological implementation details might not be as groundbreaking or novel within the recent advancements in the field of AI and LLMs. The work seems in the flow of LLM agents, while I think it is more of a LLM project rather than a technologically-solid paper.
- Although the paper demonstrates the effectiveness of the D2A agent in a specific textual home environment, there might be questions about its generalization and scalability to other environments or domains.
- The authors have established a set of concepts, such as human needs, desires, characteristics, and values, to guide the model's behavior. Although I understand the authors' intent to direct the generation of human-like daily activities, I am uncertain about the definition and composition of these intermediate variables. They are determined by the authors without sufficient psychological support or experimental validation to substantiate the overall design. How is the overall pipeline designed? I.e., for the five-level Maslow model, why you further define 11 desire dimensions?
- The experiment is limited. The testing environment is confined to a single room containing a kitchen, living area, bedroom, and bathroom. It is unclear whether there is any randomness or variation between each epoch's setup, such as the rooms in the house or the items within the room. The experiments should be conducted in a wider variety of settings to minimize the impact of environmental bias and to get general ideas. How many different scenes or settings are included in the experiment? Also, for LLMs, have the authors studied how the prompt design will influence the overall results?

**Questions:**

See the Weaknesses.

---

> ### Author Response · Authors · 2024-11-25
> **Response to Reviewer UXrD (1)**
>
> Thank you for your detailed and thoughtful review. We appreciate your recognition of the originality of our framework, the quality of our writing, and the comprehensive comparative analysis we conducted. Your feedback highlights key areas for improvement, and we have carefully considered each point to enhance our work. Below, we address your concerns and questions in detail.
>
> >**W1**: Limited technological innovation.
>
> **R1**: We sincerely appreciate your recognition of the conceptual and theoretical innovation behind our desire-driven autonomy framework. However, we respectfully hold a different perspective regarding the concern of `Limited Technological Innovation`. We believe that research capable of **driving paradigm shifts holds far greater impact and importance than work that merely introduces incremental tricks to improve performance by a few points on specific tasks**. Our work is dedicated to fostering such a paradigm shift, aiming to demonstrate that **leveraging desire-based intrinsic motivations to model agents in open-world scenarios offers significant advantages over traditional modeling approaches**. Building on this paradigm shift, the technological innovations you mentioned gain even broader potential for development and application. These **innovations are also highly compatible with our framework**. For instance, in the components of activity proposal, evaluation, and selection, we integrated the concept of single-layer Monte Carlo Tree Search (MCTS) with the human habit of pre-imagining activity outcomes, seamlessly embedding it within our framework.  Looking ahead, we are committed to further exploring technologies that align with and enhance our intrinsic motivation framework. This effort will enable us to continue improving agent performance across diverse open-world tasks, further advancing and solidifying this paradigm shift in both theory and application.
>
> >**W2**: Generalization and scalability to other environments.
>
> **R2**: To address your concerns about the scalability of our experiments, we added a multi-agent social scenario in an outdoor park party (See Appendix `A.2` and `A.3`).  Specifically, we provided a variety of interactive facilities and introduced background agents who could engage in conversations and interactions with the tested autonomous agents. Due to time constraints, we were unable to generate a large amount of experimental data in the new environment for a reliable comparison of human likeness. However, we conducted a statistical analysis on some of the action sequences obtained from the indoor and newly added outdoor scenarios. We first visualized the activity sequences obtained from the indoor and outdoor multi-agent scenarios using word cloud plots. Next, we defined several statistical labels to categorize and count the actions generated by each agent in the outdoor scenario. The results of these two statistical experiments demonstrate that our D2A can produce **diverse and balanced** actions in both indoor and outdoor scenarios (See more details in Appendix`D`). The **consistent statistical conclusions** obtained here, along with **the reasonable activity sequences in the multi-agent environment** (See Appendix `X`), demonstrate the generalization and adaptability of our framework across different settings.

---

> ### Author Response · Authors · 2024-11-25
> **Response to Reviewer UXrD (2)**
>
> >**W3**: Psychological support.
>
> **R3**: We appreciate your thoughtful question regarding the definition and composition of the intermediate variables, such as human needs, desires, characteristics, and values, used to guide the model’s behavior. Our design is inspired by the theory of needs that `needs can drive actions`, but we consider that there is **no universally agreed-upon standard for defining the conceptual space of needs or values.** The 11 dimensions implemented in our work represent a relatively simplified design intended to strike **a balance between theoretical inspiration and computational cost**. In the newly added outdoor social scenario, we added some of the value dimensions in desire while maintaining their effectiveness in guiding action generation. Specifically, the added dimensions included [`recognition`,`sense of control`,`sense of superiority`]. This demonstrates the flexibility and adaptability of our framework in incorporating alternative designs while preserving its core functionality. In our future work, we aim to explore more generalizable and systematic models of human intrinsic motivation to further refine and enhance the realism and robustness of the framework. The simulating activities, driven by different desire models, can also benefit psychology research,  verifying the limitations of existing theories/hypotheses or exploring a more accurate understanding of the desire models.
>
>
> >**W4**: The experiment is limited (Randomness and prompt design).
>
> **R4**: We have supplemented new experiments by introducing randomness into the indoor environment. Specifically, we predefined an item list for each room, and during the environment generation process for each experimental round, a certain number of items were randomly selected from each room’s list. These items were then used as context to prompt the LLM to generate a detailed textual representation of the indoor experimental environment.
>
> We conducted experiments in these randomized environments following the same procedures as in Sections 6.3.1 and 6.3.2, and **reached consistent conclusions**: (1) D2A outperformed other baselines in GPT-4o evaluations by generating more human-like activity sequences. (2) D2A’s action sequences achieved the lowest dissatisfaction value in desire. These findings demonstrate that our framework and design are **robust in randomized environments**, with no evidence of environmental bias. More details are introduced in Appendix `Z`. At the same time, we acknowledge the value of broader experimental settings. Beyond the `outdoor social environment` we discussed earlier, we plan to introduce various new experimental settings in future work to further benchmark agents’ capabilities across diverse scenarios.
>
> As researchers and users of LLMs, we recognize the inherent difficulty in qualitatively assessing the exact impact of prompts in experiments. However, our observations highlight a significant advantage of the D2A framework: incorporating desire descriptions into the prompts effectively guides the agent’s action selection process. This approach stands in contrast to baseline methods that rely heavily on profiles or explicit goals, often resulting in overly concentrated or repetitive actions. By anchoring actions to intrinsic desires, D2A fosters greater diversity and generates more human-like activity sequences, underscoring its robustness and adaptability. More details are discussed in Appendix `D`.

---

> ### Author Response · Authors · 2024-11-27
>
> Thank you for your time and thoughtful review. Your valuable feedback has been instrumental in helping us enhance and refine our paper. We are pleased to share that the concerns raised by other reviewers have been fully addressed, and they have all raised their scores as a result. If you have any concerns or require further clarification, we would be more than happy to provide additional explanations.

---

> ### Comment · Reviewer_UXrD · 2024-11-28
>
> I greatly appreciate the authors' efforts in addressing my concerns, and I have carefully read the revised paper. I am willing to update my score based on the overall quality of the paper and the authors' efforts in revising it. However, to be frank, I still believe that the paper may not be technologically solid beyond designing chains or prompts to query or test LLMs, and it may be challenging to generalize to real-world applications, such as using the desire-driven autonomy framework to effectively understand human needs or assist in planning. Nevertheless, this critique should be directed more towards the topic itself rather than the authors, as they have worked diligently on this topic and provided a substantial contribution.

---

> > ### Author Response · Authors · 2024-11-29
> >
> > Thank you for your appreciation of our work. We believe our work extends beyond robotics applications - as a general framework for simulating human-like agents, it holds great potential in various fields including video games and social simulations[1, 2]. We will continuously extend the desire-driven autonomy framework to be applicable across various domains in our future work. Once again, thank you for your time and your thoughtful review. Your kindness will encourage us to contribute more to this topic.
> >
> > [1] Li, N., Gao, C., Li, M., Li, Y., & Liao, Q. (2024). EconAgent: Large Language Model-Empowered Agents for Simulating Macroeconomic Activities. arXiv preprint arXiv:2310.10436.
> >
> > [2] Tessler, M. H., Bakker, M. A., Jarrett, D., Sheahan, H., Chadwick, M. J., Koster, R., ... & Summerfield, C. (2024). AI can help humans find common ground in democratic deliberation. Science, 386(6719), eadq2852.

---

### Official Review · Reviewer_TMAG · 2024-11-03

**Soundness:** 3
**Presentation:** 4
**Contribution:** 2
**Rating:** 6
**Confidence:** 3

**Summary:**

This paper is concerned with simulating realistic trajectories of human activities in household environments. The authors introduce a text-based home environment as well as an LLM-based agent whose outputs aim to be activity traces as diverse as possible.

Inspired by Maslow's theory of needs, their agent incorporates 11 desire scalars (with each a target value, variable across agents). These desires are split according to the levels of Maslow's hierarchy. The desired levels, current levels, previous activities, and other environment information is provided to the LLM-agent in a tree-of-thought framework to generate the next activity.

The authors find the generated trajectories are deemed more likely by judge LLMs, and decrease dissatisfaction compared to other LLM-agent baselines.

**Strengths:**

## Originality

The paper is quite original, as I have not seen Maslow's hierarchy of needs used in the context of a text agent.

## Quality

The paper is well-written, experiments are quite well-designed, several seeds are provided to account for variability. The figures are nice, and the main one does a good job of summarizing how the agent works. The results of the paper support the claims made in the introduction and the abstract.

## Clarity

The paper was easy to follow and the points are clear and easy to grasp.

## Significance

The paper demonstrates a recipe for creating more realistic human daily trajectory activities. I can see the type of agent developed here being useful for other applications, such as creating LLM-based non-playable characters in video games.

**Weaknesses:**

* I am not completely convinced of the end-goal of this paper, specifically, building sequences of human activities. I see the authors justifying this goal in the potential for generating data for psychological, economic or sociological academic study. However, the validity of the generated behavior with respect to at least one downstream application is not investigated in the paper. How to make sure the data generated is useful in these contexts?
* The introduction also briefly argues that building agents with behaviors aligning with human ones will guarantee their intelligence (the Turing test argument). But unfortunately this does not seem to be the case; I cannot see how giving agents simulated desires will make them score higher on GSM8K for instance.
* A human-judge evaluation of the validity of the AI judge would be nice (although I am still pretty convinced of the comparison results)
* I think there is a methodological flaw in the design of the human reference agent: the human is (as far as I understood) given a list of the desires, and the criteria is whether the agents are able to minimize dissatisfaction for those same desires. A better reference would be a dataset of real human activities on a stay-at-home day;

**Questions:**

* My foremost question would be about the practical use of generating human daily activity data. I do not know this subject, and it might be the focus of an entire subcommunity which I am not familiar with. It would be important for the authors to elaborate on this point, and I am ready to reconsider my score if the demonstration is convincing;
* What do the statistics of the generated activities look like?
* Is there a list of predetermined activities one may do? (predefined action space?)

## Notes

* I think that the naturalness of activities might also come from the fact that desires are satisfied for quite some time giving the agent the opportunity to concentrate on more diverse activities than in the baselines.

## Suggestions

* I believe this paper should cite Park et. al 2023 (https://arxiv.org/abs/2304.03442), a seminal work in human behavior simulation, and Colas et. al. 2023, which presents an intrinsically-motivated agent operating on other principles than Maslow's hierarchy (https://arxiv.org/abs/2305.12487).

---

> ### Author Response · Authors · 2024-11-25
> **Response to Reviewer TMAG (1)**
>
> We appreciate your acknowledgment of the originality of our work and the quality of the presentation. We are confident that the following responses will effectively address the concerns raised in your comments.
>
> >**W1 & Q1**: End-goal of the paper.
>
> **R1**: Thank you for raising the question regarding the importance of simulating and generating human activity sequences. In our paper, we only listed a few applications that have been widely recognized in academia, such as their use in psychology, economics, and sociology research. However, this **does not mean the application scope is limited to these fields.** On the contrary, the task of simulating human activity's significance and value lies in its **immense potential for real-world applications.** We fully agree with **Reviewer gp3a’s statement**:
> *By adopting an intrinsic motivation model, this framework could be used for applications such as social robotics, interactive gaming, and assistive technology, where human-like adaptability and engagement are essential.*
> It may appear that we are merely generating human-like activity sequences; indeed, this task can reflect how well agents behave within a specific environment——Our ultimate objective is to **equip agents/robots with the intrinsic motivation** we have designed so they can **act more rationally and intelligently in real-world scenarios**. When applied to assistive technology, this approach may have the potential to enable service agents to **better align with human needs and values.**
>
> >**W2**: Definition of agents' intelligence.
>
> **R2**: The discussion above can also serve as our response to your point: “I cannot see how giving agents simulated desires will make them score higher on GSM8K for instance.” We contend that **intelligent agents should not be viewed solely as `math problem solvers`.** **The manner in which their cognitive processes are modeled and their capacity to robustly select actions is critical** yet often challenging to quantify through conventional benchmarks. We argue that these capabilities are often overlooked yet essential indicators of an agent’s intelligence, particularly for general-purpose AI agents.
>
> >**W3**: A lack of Human-judge evaluation.
>
> **R3**: We appreciate your suggestion to incorporate human-judge evaluations for more convincing results. However, human evaluations are costly and challenging to scale and reproduce, which is why we use GPT-4o for evaluation. To validate the reliability of the GPT-4o evaluation, we additionally conducted a human validation study. We randomly selected 19 pairs of activity sequences for pairwise comparison (16 pairs from the indoor scenarios and 3 pairs from the newly added outdoor scenarios) and created three questionnaires, each assigned to 8 human annotators. The human annotators were asked to judge which sequence in each pair appeared more human-like. Based on the level of agreement among the annotators, we categorized the 19 samples into three groups:
>
> - **High consensus**: Over 75% agreement on one side—indicating strong human consensus.
> - **Moderate preference**: Agreement between 50.1%–74.9%—indicating moderate human preference.
> - **Difficult to distinguish**: Exactly 50% agreement—indicating humans found it difficult to decide between the two sequences.
>
> We also provided the same pairs to GPT-4o, using our evaluation prompt and both sequences as context, asking it to determine which sequence appeared more human-like. The consistency rate between human evaluations and GPT-4o’s assessments is summarized in the table below:
>
> | **Human Annotator Agreement**       | **Proportion** | **GPT-4o Consistency Rate (%)** |
> |-------------------------------------|----------------|----------------------------------|
> | High consensus ($>75\%$ agreement) | 11/19          | 100                              |
> | Moderate preference ($50.1\%$–$74.9\%$) | 5/19           | 80.0                             |
> | Difficult to distinguish ($50\%$ agreement) | 3/19           | 66.7                             |
>
> Note that for the three pairs in the table where humans found it difficult to distinguish, GPT-4o also outputs a `draw` result for two of them. This verification demonstrates the robustness and generalization of our evaluation framework and **confirms that GPT-4o’s assessments are reliable and align closely with human judgments.**

---

> ### Author Response · Authors · 2024-11-25
> **Response to Reviewer TMAG (2)**
>
> >**W4**: Methodological flaw in the design of the human reference agent
>
> **R4**: We have to point out that the `methodological flaw` mentioned in your fourth weakness stems from **a misunderstanding of our experimental design**. As described in Section 6.3.2: *“The annotators were only provided with the environment description and asked to propose activities for each time step, as if they were living in the house.”* We did not provide the annotators with the agent’s current desire states for each dimension, as we intended to collect data resembling real-life actions taken after waking up in the morning. Moreover, it is also difficult to collect data about real human activities at scale in various scenarios for reference. We chose not to directly use real human activity datasets to maintain consistency in the environment.
>
> >**Q2**: Statistics of the generated activities.
>
> **R5**: Thanks for your suggestions. To give you a more intuitive understanding of the statistical distribution of activities generated by D2A and other baselines, we first visualized the activity sequences obtained from the indoor and outdoor multi-agent scenarios using word cloud plots. Next, we defined several statistical labels to categorize and count the actions generated by each agent in the outdoor scenario. The results of these two statistical experiments demonstrate that our D2A can produce **diverse and balanced** actions in both indoor and outdoor scenarios. More details in `Appendix D`.
>
> >**Q3**: Predetermined action space
>
> **R6**: Finally, we would like to clarify that we did not provide the agents with a specific list of predetermined activities. Instead, the agents are prompted to interact with items available in the environment (strictly monitored by the Game Master). Therefore, if there is any restriction on the action space, it lies in the text-based description of the environment we provided.
>
> > **Others**: I believe this paper should cite...
>
> **R7**: Thank you for pointing out the articles we should reference. We have included them in the updated revision.

---

> > ### Comment · Reviewer_TMAG · 2024-11-26
> >
> > My concerns have been appropriately addressed, and the new multi-agent Central Park experiment makes the paper stronger. I am thus happy to raise my score.

---

> > > ### Author Response · Authors · 2024-11-27
> > >
> > > Thank you for your positive feedback. We’re delighted that our revisions have addressed your concerns and that the addition of the multi-agent Central Park experiment has strengthened the paper. We greatly appreciate your time and thoughtful review. Your encouragement inspires us to continue contributing to the field.

---

### Official Review · Reviewer_gp3a · 2024-11-06

**Soundness:** 2
**Presentation:** 2
**Contribution:** 2
**Rating:** 6
**Confidence:** 2

**Summary:**

The paper introduces a novel framework, the Desire-driven Autonomous Agent (D2A), designed to simulate human-like behaviour in daily activities. Traditional task-oriented AI agents primarily operate based on specific instructions or external rewards, but this approach often limits their ability to display intrinsic motivations similar to humans. This paper proposes an alternative: a motivation system inspired by Maslow’s hierarchy of needs, enabling the agent to autonomously generate and select activities based on desires like social interaction, self-care, and personal fulfilment. The D2A framework uses a system of 11 dimensions representing different human desires. The agent evaluates its current state and intrinsic motivations to decide on activities that align with its desires, generating more coherent and varied behaviours compared to existing models. The study uses Concordia, a text-based simulator, where a Game Master provides the environmental context for D2A to interact in a simulated household environment. The results suggest that D2A successfully simulates human-like activities by aligning with intrinsic motivations, which opens new avenues for developing desire-driven agents in various applications.

**Strengths:**

Originality: The paper presents a novel approach to simulating human-like daily activities using a desire-driven framework inspired by Maslow’s hierarchy of needs. Unlike traditional AI agents that rely on specific instructions or task-based rewards, the Desire-driven Autonomous Agent (D2A) framework introduces intrinsic motivation as the driving factor. This approach is unique in that it models a human-like motivational system, enabling the agent to select actions autonomously based on intrinsic desires rather than predefined goals.

Quality: The paper is good quality with descriptive figures and clear results.

Clarity: The paper is well structured with a clear distinction provided between their proposed method and past methods and how theirs performs better.

Significance: This work holds significance for fields focused on human-like AI, agent-based simulations, and real-world applications requiring adaptive behaviour. By adopting an intrinsic motivation model, this framework could be used for applications such as social robotics, interactive gaming, and assistive technology, where human-like adaptability and engagement are essential.

**Weaknesses:**

There are a few weaknesses.
In Section 3, Problem Formulation, the math is not very clear, and it is also not explained in more detail how the activity distribution could be generated.

In Section 6.3.1, Naturalness, Coherence and Plausibility are used to evaluate the activity sequences, but these three dimensions seem to have been picked arbitrarily and I am not sure if they are enough to rigorously test the outputs.

Evaluation is done using GPT-4o but how are we to ensure that these evaluations can be taken at face value. I think the paper would do well to do a human verification of these evaluations and how reliable they are.

**Questions:**

Listed in the weaknesses section itself.

- Make section 3 more clear by adding more details.
- Give better reasoning for using the 3 dimensions of naturalness, coherence and plausibility.
- measure the reliability of GPT-4o evaluations.

---

> ### Author Response · Authors · 2024-11-25
> **Response to Reviewer gp3a**
>
> Thank you for your thoughtful and detailed review of our work. We appreciate your recognition of the originality and significance of the Desire-driven Autonomous Agent (D2A) framework, as well as your positive feedback on the structure and clarity of the paper. Below, we address the weaknesses and questions you raised:
>
> > **W1&Q1**: Unclear mathematics in Section 3.
>
> **R1**: We apologize for any confusion caused by the mathematical expressions in our paper, which may not have clearly conveyed the step-by-step generation process. We have made efforts to clarify the mathematical expressions in the revised version of the paper. The formulas in our paper **are not strict mathematical formulas or probability distributions** but rather a way to **illustrate the context or information** we provide to the agent and the types of outputs that can be sampled based on that context. Specifically, in `Section 3`, we aim to uniformly denote the activity generation process as $a_t \sim Agent ( \cdot | a_{0:t-1}, o_{0:t-1}, I, p, e; \theta)$ , which represents generating the activity at the  $t$-th step given past activities $a_{0:t-1}$, observations $o_{0:t-1}$, customized information $I$, profile $p$, and environment $e$. We have elaborated on the specific steps involved in D2A’s activity generation process in the later sections. However, due to space constraints, we could not detail the specific steps for all the baseline methods. You can refer to the prompts provided in the appendix for further understanding.
>
> > **W2&Q2**: Reasoning behind the use of the three evaluation dimensions: naturalness, coherence, and plausibility.
>
> **R2**: Regarding your concerns about the three evaluation criteria—**Naturalness, Coherence, and Plausibility**—we would like to clarify their basis and rationale. These criteria were developed with reference to *Thagard, P. (2002). Coherence in Thought and Action, MIT Press*. According to Thagard’s framework, evaluating whether an action sequence appears human-like necessitates examining its alignment with typical human patterns. Specifically, we considered:
>
> - **Naturalness:** To assess whether the actions conform to typical human abilities and behaviors, avoiding repetitive or physically implausible actions.
> - **Coherence:** To ensure logical and sequential consistency in the flow of actions, avoiding abrupt transitions or contradictions.
> - **Plausibility:** To confirm that actions do not deviate excessively from what would be regarded as normal human behavior.
>
> These three dimensions collectively address critical aspects of evaluating human-like behavior pattern in action sequences. While we acknowledge that other methods may exist, we believe these dimensions provide a comprehensive and practical approach for assessing human-likeness. We have included the specific prompts provided to GPT-4o's evaluation in the `Appendix Y` for your review.
>
>
> > **W3&Q3**: Reliability of GPT-4o's evaluation.
>
> **R3**: Regarding the reliability of GPT-4o’s evaluations, we additionally conducted a **human verification experiment** to validate the consistency between GPT-4o and human judges. To be specific, we randomly selected 19 pairs of activity sequences for pairwise comparison (16 pairs from the indoor scenarios and 3 pairs from the newly added outdoor scenarios) and created three questionnaires, each assigned to 8 human annotators. The human annotators were asked to judge which sequence in each pair appeared more human-like. Based on the level of agreement among the annotators, we categorized the 19 samples into three groups:
>
> - **High consensus**: Over 75% agreement on one side—indicating strong human consensus.
> - **Moderate preference**: Agreement between 50.1%–74.9%—indicating moderate human preference.
> - **Difficult to distinguish**: Exactly 50% agreement—indicating humans found it difficult to decide between the two sequences.
>
> We also provided the same pairs to GPT-4o, using our evaluation prompt and both sequences as context, asking it to determine which sequence appeared more human-like. The consistency rate between human evaluations and GPT-4o’s assessments is summarized in the table below:
>
> | **Human Annotator Agreement**       | **Proportion** | **GPT-4o Consistency Rate (%)** |
> |-------------------------------------|----------------|----------------------------------|
> | High consensus ($>75\%$ agreement) | 11/19          | 100                              |
> | Moderate preference ($50.1\%$–$74.9\%$) | 5/19           | 80.0                             |
> | Difficult to distinguish ($50\%$ agreement) | 3/19           | 66.7                             |
>
> Note that for the three pairs in the table where humans found it difficult to distinguish, GPT-4o also outputs a `draw` result for two of them.
> This verification demonstrates the robustness of our evaluation framework and **confirms that GPT-4o’s assessments are reliable and align closely with human judgments.**

---

> > ### Comment · Reviewer_gp3a · 2024-11-26
> >
> > Thank you for addressing my concerns. I have raised my score accordingly.

---

> > > ### Author Response · Authors · 2024-11-27
> > >
> > > We appreciate your valuable feedback on the review and we believe that addressing reviewers' concerns thoroughly is crucial for improving our work.
> > >
> > > Thank you again for your time and your thoughtful review. Your kindness and insights will encourage us to continuously contribute to the field.

---

### Meta-Review · Area_Chair_FSjT · 2024-12-19

**Metareview:**

**Summary**: This paper introduces the Desire-Driven Autonomous Agent (D2A), a framework inspired by Maslow’s hierarchy of needs, to simulate human-like daily activities in text-based environments. By leveraging a motivational framework encompassing 11 dimensions of human-like desires (e.g., hunger, social connection, self-care), D2A autonomously generates and selects activities to fulfill these desires. Experimental results, evaluated using metrics like naturalness, coherence, and plausibility, demonstrate that D2A outperforms existing baselines such as ReAct and BabyAGI in generating plausible and coherent activity sequences. The authors also highlight potential applications in social robotics, interactive gaming, and assistive technology.

**Strengths**:
- Originality: Most reviewers commented on the originality of simulating human-like daily activities using a desire-driven framework inspired by Maslow’s hierarchy of needs.
- Clarity: Reviewers found the paper easy to follow, well-written, and clear.
- Thorough evaluation: Reviewers found experiments to be well-structured, with comparisons to relevant baselines and a variety of evaluations using both quantitative metrics and qualitative assessments. The addition of human annotator validation during the rebuttal strengthened the evaluation framework.

**Weaknesses**:
- Limited technological innovation: Reviewer `UXrD` pointed out that the paper is more conceptually interesting but not so much technologically interesting. I don't necessarily have a problem with conceptually novel but not technologically novel papers, however I would expect the conceptual innovation to lead to considerable empirical benefits. Unfortunately, looking at the results figures in the paper, the gains over baselines seem modest, especially with the large overlapping error bars.
- Generalization/Scalability concerns: Although the paper demonstrates the effectiveness of the D2A agent in a specific textual home environment, it's unclear whether it would generalize and scale to more realistic environments or domains, which undermines the practical relevance of the results. Moreover, the framework relies on fixed desire dimensions and mappings, which even further limits its adaptability to dynamic or unpredictable environments. Reviewers noted this as a potential constraint on the method's scalability. I'd expect the next iteration of this manuscript to have some discussion on this point.
- Evaluation reliance on GPT-4o: While the authors conducted a limited human annotator study, the heavy reliance on GPT-4o for evaluation metrics raised concerns about the robustness and reliability of the results. Ideally this paper would have extended human annotator study results, but I appreciated even the limited results during the rebuttal.

**Recommendation**: This paper presents a compelling and original framework for simulating human-like activities. Despite the listed weaknesses, in particular the limited generalization and scalability and the modest empirical gains, I still think there is merit to the conceptual novelty of this paper. I vote Accept for poster.

**Additional Comments On Reviewer Discussion:**

- The authors supplemented GPT-4o evaluations with human annotator validation, which helped mitigate concerns but did not fully resolve doubts about the over-reliance on automated metrics. More comprehensive human evals would be ideal, but this was a good step.
- The framework’s originality and potential applications were praised, but methodological concerns about evaluation and generalizability prevented stronger support for acceptance. Authors should provide extended discussion on this point in the final manuscript.

---

### Decision · Program_Chairs · 2025-01-22

Accept (Poster)